# Safeguarded Stochastic Polyak Step Sizes for Non-smooth Optimization: Robust Performance Without Small (Sub)Gradients

Dimitris Oikonomou [1 2]   Nicolas Loizou [1 3]

## Abstract

The stochastic Polyak step size (SPS) has proven to be a promising choice for stochastic gradient descent (SGD), delivering competitive performance relative to state-of-the-art methods on smooth convex and non-convex optimization problems, including deep neural network training. However, extensions of this approach to non-smooth settings remain in their early stages, often relying on interpolation assumptions or requiring knowledge of the optimal solution. In this work, we propose a novel SPS variant, Safeguarded SPS ($\text{SPS}_{safe}$), for the stochastic subgradient method, and provide rigorous convergence guarantees for non-smooth convex optimization with no need for strong assumptions. We further incorporate momentum into the update rule, yielding equally tight theoretical results. Comprehensive experiments on convex benchmarks and deep neural networks corroborate our theory: the proposed step size achieves competitive performance to existing adaptive baselines and exhibits stable behavior across a wide range of problem settings. Finally, in the context of deep neural network training, the gradient norms under our step size do not collapse to (near) zero, indicating robustness to vanishing gradients.

## 1. Introduction

Adaptive optimization methods have become fundamental tools in machine learning, offering robustness and eliminating the need for manual learning rate tuning. Among the most prominent are AdaGrad and Adam. AdaGrad (Duchi et al., 2011; Choudhury et al., 2024) adapts the learning rate individually for each parameter by accumulating past squared gradients, making it well-suited for sparse features but often suffering from decreasing learning rates over time. Adam (Kingma & Ba, 2015) extends this idea by maintaining exponential moving averages of both the gradient and its squared values, and correcting for initialization bias. As a result, Adam has demonstrated strong empirical performance and has become a standard optimizer in deep learning applications due to its stability and ease of use, (Vaswani et al., 2017; Guo et al., 2022; Peebles & Xie, 2023).

In a different direction, adaptive optimization algorithms using Polyak-type step sizes have started gaining recognition for their simplicity and strong practical performance. Unlike traditional adaptive methods that rely solely on gradient information, these approaches determine the learning rate using function values. The classical Polyak step size (PS), originally introduced by Polyak (1987), was proposed as an efficient rule for step size selection in gradient descent for solving convex optimization problems. Although rooted in early optimization literature, these ideas have recently seen a resurgence, particularly in machine learning applications. The Polyak step size was recently extended to stochastic settings. Loizou et al. (2021) proposed and analyzed stochastic gradient descent (SGD) with Stochastic Polyak Step size (SPS) and demonstrated convergence guarantees for convex and non-convex problems while retaining the simplicity of the original rule. The proposed SPS comes with strong convergence guarantees and competitive performance in training DNNs, and it is particularly useful when training over-parameterized models (Loizou et al., 2021). In the last few years, many other works have explored the use of stochastic Polyak step sizes in different training algorithms, including SGD (Garrigos et al., 2023; Orvieto et al., 2022; Jiang & Stich, 2023; Gower et al., 2021; Orabona & D'Orazio, 2025), Stochastic Mirror Descent (D'Orazio et al., 2023), Local SGD (Mukherjee et al., 2024), and SGD with Momentum (Wang et al., 2023; Schaipp et al., 2024; Oikonomou & Loizou, 2025; Gower et al., 2025). Nevertheless, nearly all existing analyses assume *convexity and smoothness* while robust guarantees in the non-smooth regime remain scarce (Orabona & D'Orazio, 2025). The

[1]Mathematical Institute for Data Science (MINDS), Johns Hopkins University, Baltimore, MD, USA [2]Department of Computer Science, Johns Hopkins University, Baltimore, MD, USA [3]Department of Applied Mathematics and Statistics, Johns Hopkins University, Baltimore, MD, USA. Correspondence to: Dimitris Oikonomou, Nicolas Loizou <doikono1@jh.edu, nloizou@jhu.edu>.

*Proceedings of the $43^{rd}$ International Conference on Machine Learning*, Seoul, South Korea. PMLR 306, 2026. Copyright 2026 by the author(s).

understanding of practical and efficient stochastic Polyak-type step sizes in the arguably more challenging non-smooth regime is precisely the main focus of our work.

**Problem Setup and Main Algorithms.** We focus on the unconstrained finite–sum optimization problem

$$\min_{x \in \mathbb{R}^d} \left[ f(x) = \frac{1}{n} \sum_{i=1}^{n} f_i(x) \right], \qquad (1)$$

where each component function $f_i : \mathbb{R}^d \rightarrow \mathbb{R}$ is *convex, Lipschitz and **non-smooth*** as well as lower bounded by $\ell_i^*$. Let $X^*$ denote the set of minimizers of (1). We assume that $X^* \neq \emptyset$. This problem is the cornerstone of many machine learning tasks, (Hastie et al., 2009), where the vector $x$ represents the model parameters, $f_i(x)$ is the loss related to the training point $i$, and the goal is to minimize the empirical risk $f(x)$ across all training points.

Two widely used algorithms for solving stochastic, non-smooth convex optimization problems of the form (1) are (i) the *Stochastic Subgradient Method* (SSM) and (ii) the more recent *Iterate Moving Average* (IMA), an equivalent algorithm to the stochastic subgradient method with momentum (Sebbouh et al., 2021).

SSM, tracing back to the seminal work of Robbins & Monro (1951) and later formalized for convex objectives by Shor (1985) and Nedić & Bertsekas (2001). It has the following update rule

$$x^{t+1} = x^t - \gamma_t g_i^t, \qquad (SSM)$$

where $g_i^t \in \partial f_i(x^t)$ is the stochastic subgradient, $i$ is uniformly drawn from $\{1, \dots, n\}$, and a $\gamma_t > 0$ is the step size of the method.

IMA extends SSM by adding a new iterate $(z^t)$: Each iteration first performs a subgradient step on $z^t$ and then averages the result with the previous iterate $x^t$. The update rule is given by:

$$z^{t+1} = z^t - \eta_t g_i^t, \qquad \text{where} \qquad g_i^t \in \partial f_i(x^t)$$
$$x^{t+1} = \frac{\lambda_{t+1}}{\lambda_{t+1}+1} x^t + \frac{1}{\lambda_{t+1}+1} z^{t+1}. \qquad (IMA)$$

Defazio & Gower (2021) shows that this two-sequence scheme is *algebraically equivalent* to the more familiar Stochastic Heavy Ball (SHB) momentum update rule $x^{t+1} = x^t - \hat{\gamma}_t g_i^t + \beta(x^t - x^{t-1})$ with $1 + \lambda_{t+1} = \lambda_t \beta_t$ and $\eta_t = (1 + \lambda_{t+1})\hat{\gamma}_t$, (Ma & Yarats, 2019; Kidambi et al., 2018; Liu et al., 2020; Sebbouh et al., 2021; Oikonomou & Loizou, 2025).

In our work, we focus on adaptive variants of both SSM and IMA, and provide Polyak-type step-sizes $\gamma_t$ for solving convex non-smooth optimization problems.

**Prior non-smooth Polyak-type Results.** For the Stochastic Subgradient Method (SSM) on convex and Lipschitz objectives, Loizou et al. (2021) proved that SSM with the step size $\gamma_t = \frac{f_i(x^t) - f_i^*}{\|g_i^t\|^2}$, where $f_i^* = \inf_{x \in \mathbb{R}^d} f_i(x)$, converges at a rate of $\mathcal{O}(T^{-1/2})$ *only* under the strong *interpolation* assumption (i.e. there exists $x^* \in X^*$ such that $f_i(x^*) = f_i^*$ for all $i$). Garrigos et al. (2023) later proposed the step size

$$\gamma_t = \frac{[f_i(x^t) - f_i(x^*)]_+}{\|g_i^t\|^2}, \qquad (SPS^*)$$

where $[z]_+ = \max\{z, 0\}$, obtaining the same $\mathcal{O}(T^{-1/2})$ bound without interpolation, but at the cost of requiring knowledge of each individual optimal loss value (i.e., $f_i(x^*)$).

In the momentum setting, Gower et al. (2025) extended this idea to the Iterate Moving Average (IMA) update rule, showing that the IMA method with the step size

$$\eta_t = \frac{[f_i(x^t) - f_i(x^*) + \lambda_t \langle g_i^t, x^t - x^{t-1} \rangle]_+}{\|g_i^t\|^2},$$
$$(IMA\text{-}SPS)$$

achieves the same rate for both the Cesàro average and the last iterate, again assuming access to $f_i(x^*)$. For a summary of these results, we refer to Table 1.

More recently, the concurrent work Orabona & D'Orazio (2025), proves that several Polyak-type step-sizes for SSM can be interpreted as a gradient descent in a carefully selected surrogate loss. As a special case of their approach, they proved that the original $SPS_{max}$ update rule of Loizou et al. (2021), $\gamma_t = \min\{\frac{f_i(x^t) - f_i^*}{\|g_i^t\|^2}, \gamma_b\}$ where $\gamma_b > 0$, works without any modification in the non-smooth regime. As we explain in Section 2.2, even if $SPS_{max}$ comes with nice convergence guarantees for different classes of problems through the approach of Orabona & D'Orazio (2025), this rule could be ineffective in ML tasks without the smoothing trick of the upper bound $\gamma_b$ of Loizou et al. (2021). In practice, $\gamma_b$ is actually selected in the majority of iterations, making the impact of the actual Polyak step-size selection unclear. On the other hand, our proposed *Safeguarded* Polyak step sizes do not suffer from this issue; by construction, they avoid the use of a bound on the step-size selection and instead use a bound on the norm of the gradients. The results of Orabona & D'Orazio (2025) hold only for SSM and cannot trivially be extended to the momentum variants (IMA), which we also handle in our work.

**Limitations and our remedy.** Taken together, the existing variants of Polyak-type algorithms either (i) converge only under interpolation, or (ii) rely on oracle information such as $f_i(x^*)$ that is unavailable in practice or (iii) in practice heavily depend on the selection of the upper bound $\gamma_b$.

| Work | Step Size | No Interpolation? | No $f_i(x^*)$? | Rate |
|------|-----------|:-----------------:|:--------------:|:----:|
| *Stochastic Subgradient Method* | | | | |
| Loizou et al. (2021) | $\gamma_t = \frac{f_i(x^t)-f_i^*}{\|g_i^t\|^2}$ | ✗ | ✓ | $\mathcal{O}(1/\sqrt{T})$ |
| Garrigos et al. (2023) | $\gamma_t = \frac{[f_i(x^t)-f_i(x^*)]_+}{\|g_i^t\|^2}$ | ✓ | ✗ | $\mathcal{O}(1/\sqrt{T})$ |
| Theorem 3.1 | $\gamma_t = \frac{f_i(x^t)-\ell_i^*}{\max\{\|g_i^t\|^2,M\}}$ | ✓ | ✓ | $\mathcal{O}(1/\sqrt{T}+\sigma^2)$ |
| *IMA (Momentum)* | | | | |
| Gower et al. (2025) | $\eta_t = \frac{[f_i(x^t)-f_i(x^*)+\lambda_t\langle g_i^t, x^t-x^{t-1}\rangle]_+}{\|g_i^t\|^2}$ | ✓ | ✗ | $\mathcal{O}(1/\sqrt{T})$ |
| Theorem 3.5, Theorem 3.6 | $\eta_t = \frac{[f_i(x^t)-\ell_i^*+\lambda_t\langle g_i^t, x^t-x^{t-1}\rangle]_+}{\max\{\|g_i^t\|^2,M\}}$ | ✓ | ✓ | $\mathcal{O}(1/\sqrt{T}+\sigma^2)$ |

*Table 1.* Overview of methods with Polyak-type step sizes analyzed in the convex non-smooth stochastic setting. For every method we list the explicit rule, indicate whether the theory (i) holds *without* the interpolation assumption and (ii) avoids the oracle values $f_i(x^*)$, and report the proven convergence rate on convex–Lipschitz objectives. Rows shaded in green are the new contributions of this work. The constant $M$ in our step sizes is the safeguard constant, for more details see Section 2 and the variance $\sigma^2$ is defined in (4).

The step sizes we introduce in this work eliminate these drawbacks: they require only a lower bound and a single safeguard parameter $M$, and match the $\mathcal{O}(T^{-1/2})$ rate for convex, Lipschitz objectives (up to a neighborhood), in both plain SSM and its momentum variant IMA.

### 1.1. Main Contributions

Our main contributions are summarized below:

◇ **Safeguarded Polyak Step size for SSM.** We design a new Polyak-type rule, named safeguarded SPS ($\text{SPS}_{safe}$) for SSM, that prevents the denominator of the stochastic Polyak step size from becoming small. This single safeguard removes the need for any oracle information (e.g. $f_i(x^*)$) and attains $\mathcal{O}(T^{-1/2})$ convergence to a neighborhood of the solution, for stochastic, convex, and Lipschitz objectives *without* the interpolation assumption.

◇ **An in-depth understanding of $\text{SPS}_{safe}$.** We explain the benefits of $\text{SPS}_{safe}$ in terms of theory and experiments compared to prior works on Polyak-type step sizes. The proposed rule is, to our knowledge, the first Polyak-type step size for SSM that remains is *not hard-capped by a constant step-size ceiling*: it never reduces to a constant update, irrespective of the chosen safeguard threshold. Earlier variants (Loizou et al., 2021; Wang et al., 2023; Zhang et al., 2025) can yield a *constant* step size once a user-specified upper bound on the step size is small enough. In addition, we establish a connection between the proposed step size rule and the clipping mechanism used extensively in modern DNN training. The safeguarding mechanism can be interpreted as an *in-step* gradient-clipping operation, thereby supplying the first theoretical guarantees for Polyak-style *clipped* SSM, an update rule widely used to stabilise training in DNNs.

◇ **Safeguarded Polyak-type step size for IMA and**

last-iterate convergence. We extend the safeguarded step size ideas to the Iterate Moving Average (IMA) update rule, yielding the Safeguarded Polyak-type step size IMA-$\text{SPS}_{safe}$. For this step size selection, we prove $\mathcal{O}(T^{-1/2})$ convergence for IMA in terms of *both* the Cesàro average and the last-iterate. Our proposed analysis provides the first convergence guarantees for an adaptive momentum method (through the equivalence of IMA and SSM with heavy ball momentum) that does not require any strong assumption (e.g., the knowledge of $f_i(x^*)$, (Gower et al., 2025)).

◇ **Numerical Evaluation.** In Section 4, we present an extensive empirical study of our step size, including sensitivity analysis of our step size and comparison with other Polyak step sizes in the non-smooth setting. We also assess the performance of $\text{SPS}_{safe}$ and IMA-$\text{SPS}_{safe}$ in training deep neural networks for multi-class image classification problems and track the gradient norms under $\text{SPS}_{safe}$, observing that they remain larger than those of the smoothed Polyak baseline.

## 2. Safeguarded Stochastic Polyak Step sizes

This section introduces the *Safeguarded* Polyak step sizes for SSM and IMA and we explain how our theory differs from previous works. Next, we show how the safeguarded step sizes improve practical behaviour in deep-network training. Finally, we show that the safeguard can be interpreted as an adaptive form of gradient clipping, thereby combining Polyak updates with clipped-SSM techniques.

**$\text{SPS}_{safe}$ for the Stochastic Subgradient Method.** To stabilise Polyak step sizes in the non-smooth setting we introduce

$$\gamma_t = \frac{f_i(x^t)-\ell_i^*}{\max\{\|g_i^t\|^2, M\}}, \qquad (\text{SPS}_{safe})$$

where $g_i^t \in \partial f_i(x^t)$ and $M > 0$ is a user–chosen safeguard. The $\max\{\cdot, M\}$ term prevents the denominator from approaching zero, thereby avoiding the *exploding step size* problem that arises when $\|g_i^t\| \to 0$ in deep neural networks. Earlier work of Loizou et al. (2021) controlled the same phenomenon by clipping the *whole* polyak step size, taking $\gamma_t = \min\{\text{Polyak step}, \gamma_b\}$ with a fixed upper bound $\gamma_b$. By contrast, $\text{SPS}_{safe}$ keeps the numerator intact and instead caps the denominator from below, preventing the reciprocal $1/\|g_i^t\|^2$ from blowing up when $\|g_i^t\| \to 0$; this empirically produces smoother step sizes and tight theoretical bounds (see Theorem 3.1).

In essence, the single hyper-parameter $M$ replaces both the clipping constant $\gamma_b$ of $\text{SPS}_{\max}$ and the oracle values $f_i(x^*)$ required by $\text{SPS}^*$/IMA-SPS, yielding an adaptive step-size family for non-smooth optimization that requires only a lower bound and a safeguard parameter $M$.

**IMA-SPS$_{safe}$ for momentum.** The Iterate-Moving-Average (IMA) framework of Gower et al. (2025) selects $\eta_t = [f_i(x^t) - f_i(x^*) + \lambda_t\langle g_i^t, x^t - x^{t-1}\rangle]_+/\|g_i^t\|^2$, but requires the unknown optimal loss $f_i(x^*)$. We remove this oracle dependence and simultaneously safeguard against vanishing gradients with

$$\eta_t = \frac{[f_i(x^t) - \ell_i^* + \lambda_t\langle g_i^t, x^t - x^{t-1}\rangle]_+}{\max\{\|g_i^t\|^2, M\}},$$
(IMA-SPS$_{safe}$)

where $\lambda_t \geq 0$ is the usual IMA momentum parameter. When $\lambda_t = 0$, this reduces to $\text{SPS}_{safe}$, while for $\lambda_t > 0$ it becomes a safeguarded analogue of stochastic heavy ball that enjoys last-iterate and Cesàro guarantees (Theorems 3.5 and 3.6) without any knowledge of $f_i(x^*)$.

## 2.1. Theoretical Comparison with Closely Related Works

**Comparison with Prior Results on Stochastic Subgradient Method.** To the best of our knowledge, the literature contained until recently two theoretical guarantees for Polyak-type step sizes in the convex–Lipschitz regime (non-smooth regime) (Loizou et al., 2021; Garrigos et al., 2023), both of which require strong, often impractical, assumptions. More recently, in concurrent work, Orabona & D'Orazio (2025) proved that the $\text{SPS}_{\max}$ rule of Loizou et al. (2021) can also be effective in the non-smooth regime.

Let $f_i$ be convex and $G$-Lipschitz functions and let $\overline{x}^T = \frac{1}{T}\sum_{t=0}^{T-1} x^t$. Then the above three papers provide the below convergence guarantees:

- (Loizou et al., 2021): Assume that *interpolation* condition holds. Consider the iterates of SSM with the step size given by $\gamma_t = \frac{f_i(x^t) - f_i^*}{\|g_i^t\|^2}$. Then $\mathbb{E}[f(\overline{x}^T) - f(x^*)] \leq$ $\frac{G\|x^0 - x^*\|}{\sqrt{T}}$.

- (Garrigos et al., 2023): Consider the iterates of SSM with the step size given by $\gamma_t = \frac{[f_i(x^t) - f_i(x^*)]_+}{\|g_i^t\|^2}$. Then $\mathbb{E}[f(\overline{x}^T) - f(x^*)] \leq \frac{G\|x^0 - x^*\|}{\sqrt{T}}$. The use of $[z]_+$ is needed to enforce the step size to be positive.

- (Orabona & D'Orazio, 2025): Consider the iterates of SSM with the step size given by $\gamma_t = \min\left\{\frac{f_i(x^t) - f_i^*}{\|g_i^t\|^2}, \gamma_b\right\}$. Then $\mathbb{E}[f(\overline{x}^T) - f(x^*)] \leq \frac{G\|x^0 - x^*\|}{\sqrt{T}} + \frac{\|x^0 - x^*\|^2}{\gamma_b T} + 2\hat{\sigma}^2 + G\sqrt{2\gamma_b\hat{\sigma}^2}$, where $\hat{\sigma}^2 = \mathbb{E}[f_i(x^*) - f_i^*]$.

The first two papers above rely on conditions rarely met in practice: either (i) exact interpolation or (ii) full knowledge of $f_i(x^*)$ for all $i$. Note also that when interpolation is assumed, $f_i^* = f_i(x^*)$, making the step size in both papers coincide. Using our proposed safeguarded step size $\text{SPS}_{safe}$, we achieve the same $\mathcal{O}(T^{-1/2})$ convergence to a neighborhood of the solution, and the results hold *without* assuming interpolation and *without* oracle information. The final bound has the familiar smooth-setting structure: it converges to a neighborhood whose radius scales with the gradient variance and collapses to zero in the interpolated case, thereby recovering Loizou et al. (2021) as a special instance.

The concurrent work Orabona & D'Orazio (2025) managed to establish $\mathcal{O}(T^{-1/2})$ convergence to a neighborhood for stochastic, convex non-smooth objectives *without* interpolation by directly employing the $\text{SPS}_{\max}$. The $\text{SPS}_{\max}$ choice for the step-size, as we explain in the next subsection, has nice theoretical properties but could suffer in practical scenarios as the choice of the upper bound $\gamma_b$ dictates the step-size selection ($\gamma_b$ is selected in the majority of iterations). $\text{SPS}_{safe}$ by construction, avoid the use of such a bound in step-size selection and instead use what we consider a more natural bound in the norm of gradients (part of the denominator in our step-size selection).

**Comparison with Prior Results on Iterative Moving Averaging (IMA/Momentum).** The only work that managed to have a Polyak-type stepsize for IMA for solving convex non-smooth problems is Gower et al. (2025), which proposes using $\eta_t = [f_i(x^t) - f_i(x^*) + \lambda_t\langle g_i^t, x^t - x^{t-1}\rangle]_+/\|g_i^t\|^2$. This step-size selection, similar to Garrigos et al. (2023), requires a priori knowledge of the optimal loss values $f_i(x^*)$. For this step-size selection Gower et al. (2025) proved $\mathcal{O}(1/\sqrt{T})$ convergence to the exact solution for both the average point $\overline{x}^T = \frac{1}{T}\sum_{t=0}^{T-1} x^t$ and the last-iterate. In our work, we provide similar convergence guarantees (see Theorem 3.5 and Theorem 3.6) to a neighborhood of the solution without requiring the knowledge of $f_i(x^*)$.

## 2.2. Practical considerations in training of DNNs

**How often is a Polyak rule actually used?** Polyak step sizes are praised for being *adaptive*, yet in deep neural networks experiments, they can end up acting like fixed learning rates. To investigate this phenomenon we run ResNet-20 (He et al., 2016) on CIFAR-10 dataset (Krizhevsky et al., 2009). Recall, Loizou et al. (2021) proposed the following step size

$$\gamma_t^{\text{SPS}_{\max}} = \min\left\{\gamma_t^{\text{SPS}}, \gamma_b\right\}, \gamma_t^{\text{SPS}} = \frac{f_i(x^t) - \ell_i^*}{c\|g_i^t\|^2}.$$
(SPS$_{\max}$)

For SPS$_{\max}$, we swept the constant $c$ over $c \in \{0.1, 0.2, \ldots, 1.0\}$, set $\gamma_b = 1$ and $\ell_i^* = 0$, and trained for 100 epochs. The best accuracy, which reached $87.88\%$, occurred at $c = 0.4$. However, a counter revealed that in **31.8%** of the iterations the algorithm selected the constant value $\gamma_b$ rather than the "true" Polyak step $\gamma_t^{\text{SPS}}$. Thus almost one-third of the updates were effectively constant.

In order to increase performance, Loizou et al. (2021) use the following smoothing rule for the clipping hyperparameter[1] $\gamma_b^t = \tau^{b/n}\gamma_{t-1}$ with $\tau = 2$, batch size $b$, and dataset size $n$. The step size then takes the following form:

$$\gamma_t^{\text{Smooth SPS}_{\max}} = \min\left\{\gamma_t^{\text{SPS}}, \gamma_b^t\right\} \tag{2}$$
$$= \min\left\{\gamma_t^{\text{SPS}}, \tau^{b/n}\gamma_{t-1}^{\text{Smooth SPS}_{\max}}\right\}.$$
(Smooth SPS$_{\max}$)

Adopting this rule and retuning $c$ over $c \in \{0.1, 0.2, \ldots, 1.0\}$, lifts accuracy to $89.79\%$ for $c = 0.5$, but at a hidden cost: **98.45%** of the steps now use $\gamma_t = \gamma_b^t$. In practice, the method behaves almost like a decreasing learning-rate scheme, the step size is plotted in Figure 1.

**Safeguarded Polyak.** Replacing SPS$_{\max}$ with our safeguarded step SPS$_{safe}$ yields $90.39\%$ accuracy, competitive with the smoothed baseline, while *never* collapsing to a constant value. SPS$_{safe}$ also exhibits a smoothing behaviour similar to 2, see Figure 1 for a direct comparison between the two step sizes. Crucially, this behaviour is backed by explicit convergence guarantees, whereas the smoothed heuristic offers none.

## 2.3. Connections with Clipping: Adaptive Clipped SSM via SPS$_{safe}$

A convenient way to stabilise stochastic subgradient methods is to clip each individual gradient before applying the update. Given a threshold $c > 0$, define the clipping opera-

---

[1]Similar smoothing procedures have been used in Tan et al. (2016); Vaswani et al. (2019)

tor

$$\text{clip}_c(g) := \min\left\{1, \frac{c}{\|g\|}\right\} g, \tag{3}$$

which leaves gradients with $\|g\| \leq c$ unchanged and rescales larger ones to have norm exactly $c$. The *clipped* stochastic subgradient method (clipped SSM) therefore has the following update rule:

$$x^{t+1} = x^t - \tilde{\gamma}_t\text{clip}_c\left(g_i^t\right) = x^t - \tilde{\gamma}_t\min\left\{1, \frac{c}{\|g_i^t\|}\right\} g_i^t,$$
(Clipped SSM)

with step size $\tilde{\gamma}_t > 0$. By capping overly large gradients, clipping protects against the exploding-gradient spikes while gradients with $\|g\| \leq c$ are left unchanged.

The following proposition explains how SPS$_{safe}$ can be modified for Clipped SSM, thus providing an adaptive step size for Clipped SSM. For the proof we refer to the appendix.

**Proposition 2.1.** SSM with SPS$_{safe}$ and $M = c^2$ is algebraically equivalent to Clipped SSM with the *adaptive* step size $\tilde{\gamma}_t = \frac{f_i(x^t) - \ell_i^*}{c\max\{c, \|g_i^t\|\}}$.

**Small subgradients.** Gradient clipping augments SSM with a simple safety mechanism: whenever a stochastic gradient's norm exceeds a user-set bound $c$, it is rescaled to have norm exactly $c$, while gradients with $\|g\| \leq c$ are left untouched. By capping very large updates, it prevents the exploding gradient spikes that can derail optimization. In a related but distinct spirit, the safeguard in our step-size formula SPS$_{safe}$ prevents the *learning rate* from blowing up: rather than rescaling the subgradient, it keeps the reciprocal normalisation $1/\|g_i^t\|^2$ bounded when subgradients become extremely small. There is a growing deterministic literature that combines Polyak step sizes with Clipped SSM in the non-smooth setting, notably the recent analyses of Gorbunov et al. (2025) and Takezawa et al. (2024). These results, however, target $(L_0, L_1)$-smooth objectives, whereas we focus on globally Lipschitz, but otherwise non-smooth, functions and operates in the *stochastic* regime.

## 3. Convergence Analysis

This section states the convergence guarantees for SPS$_{safe}$ and IMA-SPS$_{safe}$; full proofs (and a convergence result for a time-varying safeguard $M_t$, Theorem E.1) are deferred to the appendix. Throughout, each loss $f_i$ is convex and $G$-Lipschitz.

**Variance measure.** To quantify gradient noise in the non-smooth setting we use

$$\sigma^2 := \left(\mathbb{E}_i\left[\left(f_i(x^*) - \ell_i^*\right)^2\right]\right)^{\frac{1}{2}}. \tag{4}$$

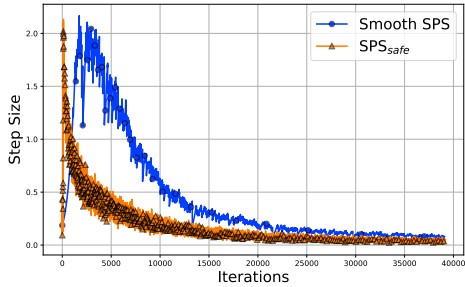 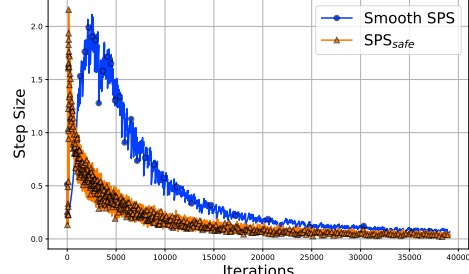

*Figure 1.* Comparison of 2 and SPS$_{safe}$ in the training of ResNet20 in CIFAR-10 (left plot) and CIFAR-100 (right plot).

The standard definition of the variance in the Stochastic Polyak step sizes literature (Loizou et al., 2021; Oikonomou & Loizou, 2025; Wang et al., 2023; Zhang et al., 2025) is given by $\hat{\sigma}^2 := \mathbb{E}_i [f_i(x^*) - \ell_i^*]$. Note that by Jensen's inequality we have that $\hat{\sigma}^2 \leq \sigma^2$. Moreover, $\sigma^2 < \infty$ whenever each $f_i$ is lower-bounded. When problem (1) is interpolated, i.e. there exists $x^* \in X^*$ such that $f_i(x^*) = f_i^*$, then choosing $\ell_i^* = f_i^*$ we get $\sigma^2 = 0$. Many modern machine learning models satisfy this condition. Examples include non-parametric regression (Liang & Rakhlin, 2020) and over-parameterized deep neural networks (Zhang et al., 2021; Ma et al., 2018).

### 3.1. Stochastic Subgradient Method

Firstly, we focus on the stochastic subgradient method, where we have the following theorem.

**Theorem 3.1.** Consider the iterates of SSM with the step size SPS$_{safe}$. Let $\overline{x}^T = \frac{1}{T} \sum_{t=0}^{T-1} x^t$. Then

$$\mathbb{E}[f(\overline{x}^T) - f(x^*)] \leq \frac{\sqrt{\max\{G^2, M\}}\|x^0 - x^*\|}{\sqrt{T}}$$
$$+ \sqrt{\frac{\max\{G^2, M\}}{M}} \sigma^2.$$

Theorem 3.1 eliminates two issues that limit the best previous bounds of Loizou et al. (2021) and Garrigos et al. (2023): it needs neither the interpolation condition nor oracle access to the values $f_i(x^*)$ and still achieves the rate $\mathcal{O}(T^{-1/2})$. Because the safeguard acts solely through the denominator, the same result, via the equivalence in Proposition 2.1, yields the first stochastic guarantee for the clipped-SSM update (Clipped SSM).

**On the $\sigma^2$-neighborhood term.** The bound of Theorem 3.1 has the familiar two-term structure of constant-step-size stochastic methods: an $\mathcal{O}(T^{-1/2})$ optimization term plus a $\sigma^2$-dependent neighborhood term. Such a neighborhood is not specific to our analysis; it is an inherent feature of any non-decreasing, non-variance-reduced stochastic method, and analogous terms appear in constant-step-size SGD (Garrigos & Gower, 2023), its momentum variants (Sebbouh

et al., 2021; Liu et al., 2020), and all prior Polyak-type analyses (Loizou et al., 2021; Oikonomou & Loizou, 2025; Orabona & D'Orazio, 2025). In general, the coefficient $\sqrt{\max\{G^2, M\}/M}$ of $\sigma^2$ is decreasing in $M$, so a larger safeguard tightens the neighborhood, a trend we confirm empirically in Section 4.1.

A natural attempt to remove the neighborhood is to decay the safeguard $M$ towards zero over time. This does not work: once $M < G^2$, the neighborhood coefficient $\sqrt{\max\{G^2, M\}/M} = \sqrt{G^2/M}$ *diverges* as $M \to 0$, so the neighborhood term blows up rather than vanishing. Hence the radius of the neighborhood cannot be driven to zero through $M$ alone. As a partial result in terms of varying $M$, Theorem E.1 in Appendix E shows that the safeguard may be replaced by a *time-varying* sequence $M_t$ without losing the $\mathcal{O}(T^{-1/2})$ rate, provided $M_t$ stays bounded away from zero and from above.

Now consider two notable specializations of Theorem 3.1. First, in the *interpolated* regime, where every sample can be fitted exactly, we choose the lower bounds $\ell_i^* = f_i^*$. The variance term then disappears and we obtain an exact convergence rate.

**Corollary 3.2** (Interpolation). Under interpolation with $\ell_i^* = f_i^*$,

$$\mathbb{E}[f(\overline{x}^T) - f(x^*)] \leq \frac{\sqrt{\max\{G^2, M\}}\|x^0 - x^*\|}{\sqrt{T}}.$$

When $M = 0$ this reproduces the step size and rate of Loizou et al. (2021), as seen in Section 2.1, thus recovering earlier results as a special case of the safeguarded framework. Next, we concentrate on the deterministic or full batch regime.

**Corollary 3.3** (Deterministic SSM). In the deterministic regime ($i = [n]$), assume $\ell^* = f^*$. Then $\sigma^2 = 0$, so Theorem 3.1 suggests

$$\min_{t \in [T]}\{f(x^t) - f(x^*)\} \leq \frac{\sqrt{\max\{G^2, M\}}\|x^0 - x^*\|}{\sqrt{T}}.$$

There is a plethora of Polyak step size analyses in the deterministic regime assuming non-smoothness. Polyak's seminal work in Polyak (1964) on deterministic subgradient descent with the Polyak step size already established an $\mathcal{O}(G\|x^0 - x^*\|/\sqrt{T})$ bound for nonsmooth convex objectives. Subsequent studies strengthened the guarantees under additional structure: Davis et al. (2018) proved *linear* convergence for the same step size when the objective is weakly convex and sharp, while Hazan & Kakade (2019) obtained an $\mathcal{O}(1/T)$ rate for strongly convex, Lipschitz functions.

### 3.2. Iterate Moving Average (Momentum)

In this section we focus on IMA. Since the Bregman divergence of a non-smooth function depends on the chosen subgradient, we first fix this choice.

**Remark 3.4** (Choice of subgradient). Throughout the momentum analysis we use the specific subgradient

$$g^t := \mathbb{E}_i[g_i^t \mid x^t],$$

the conditional expectation of the sampled stochastic subgradient $g_i^t \in \partial f_i(x^t)$. By Lemma 9.5 of Garrigos & Gower (2023), $g^t \in \partial f(x^t)$, so $g^t$ is a valid subgradient of $f$ at $x^t$. Accordingly, the Bregman divergence is taken with respect to this subgradient,

$$B_f(x^{t-1}, x^t) := f(x^{t-1}) - f(x^t) - \langle g^t, x^{t-1} - x^t \rangle \geq 0,$$

where non-negativity follows from the subgradient inequality. This is purely a notational clarification and does not affect the theorem statements or the proofs.

**Theorem 3.5.** Consider the iterates of IMA with the step size (IMA-SPS$_{safe}$) and let $(\lambda_t)_{t>0}$ be a non-increasing sequence of nonnegative reals. Then

$$\mathbb{E}[f(\overline{x}^T) - f(x^*)] + \sum_{t=0}^{T-1} \frac{\lambda_t}{T}\mathbb{E}[B_f(x^{t-1}, x^t)]$$
$$\leq \frac{\sqrt{\max\{G^2, M\}}\|x^0 - x^*\|}{\sqrt{T}} + \sqrt{\frac{\max\{G^2, M\}}{M}}\sigma^2,$$

where $\overline{x}^T = \frac{1}{T}\sum_{t=0}^{T-1} x^t$ and $B_f$ is the Bregman divergence defined in Remark 3.4.

The bound mirrors that of Theorem 3.1, but with an additional non-negative Bregman divergence term. The most common scenario for the sequence $\lambda_t$ is being held fixed ($\lambda_t = \lambda$). When $\lambda = 0$ we recover exactly Theorem 3.1. However, when $\lambda > 0$ we have an extra non-negative term on the left hand side. This suggests, but does not force, a speed-up over the plain subgradient method.

So far we have only provided guarantees for the Cesaro

average. In the next theorem we prove convergence of the last iterate.

**Theorem 3.6.** Consider the iterates of IMA with the step size (IMA-SPS$_{safe}$) and let $\lambda_t = t$. Then

$$\mathbb{E}[f(x^{T-1}) - f(x^*)] + \frac{1}{T}\sum_{t=0}^{T-1} t\mathbb{E}[B_f(x^{t-1}, x^t)]$$
$$\leq \frac{\sqrt{\max\{G^2, M\}}\|x^0 - x^*\|}{\sqrt{T}} + \sqrt{\frac{\max\{G^2, M\}}{M}}\sigma^2.$$

This result provides an explicit guarantee for the *last* iterate, often the quantity of practical interest, while retaining the same rate as the Cesàro bound. Similar remarks as in the previous section hold about the neighborhood in this regime, namely the neighborhood is decreasing with respect to safeguard $M$ (see also Section 4.1).

Fewer results exist for Polyak-type step sizes paired with momentum than for their momentum-free counterparts. Wang et al. (2023), treats a heavy-ball variant under *smooth* convex losses and achieves an $\mathcal{O}(1/T)$ rate. Oikonomou & Loizou (2025) study the Stochastic Heavy Ball momentum via IMA, again assuming smooth objectives. The only work that drops smoothness is the analysis of Gower et al. (2025), which attains the $\mathcal{O}(T^{-1/2})$ rate in the convex, Lipschitz setting, but at the cost of requiring the quantities $f_i(x^*)$. Our safeguarded momentum rule removes this dependence on $f_i(x^*)$ dependence while preserving the same rate up to a neighborhood.

## 4. Numerical Experiments

We now examine the empirical behaviour of the safeguarded step sizes SPS$_{safe}$ and IMA-SPS$_{safe}$. The first series of experiments targets convex and nonconvex *non-smooth* objectives, complementing the analysis of Section 3. The second series moves to deep-learning benchmarks, measuring the impact of the step sizes on classification accuracy. We provide the code for all of our experiments at https://github.com/dimitris-oik/sps_safe.

### 4.1. Evaluation on Non-smooth Objectives: Support Vector Machines and Phase Retrieval

In this part, we evaluate the empirical behaviour of SPS$_{safe}$ and IMA-SPS$_{safe}$. We focus on two synthetic non-smooth problems: a Support Vector Machine (SVM) and a Phase Retrieval problem. The SVM hinge loss is convex and $G$-Lipschitz and thus directly matches the assumptions of Section 3; the phase-retrieval objective is nonconvex and non-smooth, and together they cover two non-smooth regimes of interest for Polyak-type step sizes.

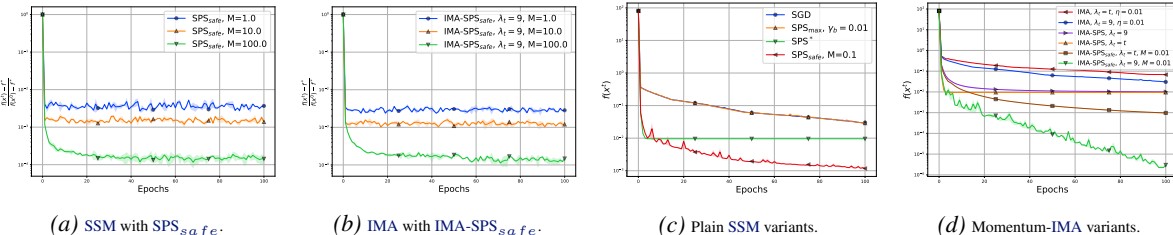

*Figure 2.* Sensitivity analysis of the safeguarded Polyak step size to the threshold $M$ (Panels a-b for Phase Retrieval) and comparison against SPS variants (Panels c–d for SVM).

**SVM.** The individual loss and subgradient are given by

$$f_i(x) = \max\left(0, 1 - b_i\langle A_i, x\rangle\right)$$
$$\partial f_i(x) = -\delta_{b_i\langle A_i, x\rangle \leq 1} b_i A_i,$$

where $\delta_X = 1$ if condition $X$ holds, and $\delta_X = 0$ otherwise.

**Phase Retrieval.** The individual loss and subgradient are given by

$$f_i(x) = \left|\langle A_i, x\rangle^2 - b_i\right|$$
$$\partial f_i(x) = 2\langle A_i, x\rangle \text{sgn}\left(\langle A_i, x\rangle^2 - b_i\right) A_i,$$

where $\text{sgn}(\cdot)$ denotes the sign function.

**Sensitivity to the safeguard $M$.** We study how the choice of the threshold $M$ influences both the plain and momentum variants of our proposed step sizes. The experiment is a synthetic phase-retrieval task with $n = 300$ samples and dimension $d = 10$; rows of $\boldsymbol{A}$ and the vector $b$ are drawn i.i.d. from $\mathcal{N}(0,1)$. We run SSM equipped with $\text{SPS}_{safe}$ and IMA equipped with $\text{IMA-SPS}_{safe}$ for 100 epochs, using a batch size of $n/10 = 30$. Three values of the safeguard are tested, $M \in \{1, 10, 100\}$, and for the momentum experiment, we set $\lambda_t = 9$, which is equivalent to the heavy-ball parameter $\beta = 0.9$. Each configuration is averaged over three independent trials; mean curves and one-standard-deviation bands are reported in Figures 2a and 2b. Both algorithms converge to progressively lower error plateaus as $M$ grows, confirming that the safeguard controls the size of the limiting neighborhood.

**Comparison with existing Polyak step sizes.** We next benchmark the safeguarded rules against their best-tuned classical counterparts. The task is a synthetic SVM with $n = 300$ samples and dimension $d = 100$; both the feature matrix $\boldsymbol{A}$ and the label vector $b$ are drawn from $\mathcal{N}(0,1)$ as in the previous section. For $\text{SPS}_{safe}$ and $\text{SPS}_{safe}$ we sweep the safeguard over $M \in \{0.01, 0.1, 1.0, 1.0, 10.0, 100.0\}$. The plain SSM is tuned over four learning rates $\gamma \in \{10^{-4}, 10^{-3}, 10^{-2}, 10^{-1}\}$, while the constant step size IMA baseline is tuned over $\eta \in \{10^{-4}, 10^{-3}, 10^{-2}, 10^{-1}\}$ for two momentum choices: $\lambda_t = 9$ (equivalent to $\beta = 0.9$) and $\lambda_t = t$. SPS* (Garrigos et al., 2023) and IMA-SPS (Gower et al., 2025) require the exact optimal values

$f_i(x^*)$. To approximate these quantities we run deterministic (full-batch) subgradient descent for $50,000$ iterations and treat the final iterate as $x^*$. All methods are trained for 100 epochs with batch size $n/10 = 30$. Every experiment is repeated three times with independent data draws; mean trajectories and one-standard-deviation bands are plotted in Figures 2c and 2d.

The safeguarded Polyak rules dominate the competition on this non-smooth problem. In the plain-SSM setting (left panel) the best tuned $\text{SPS}_{safe}$ consistently outperforms both tuned fixed-step size SSM and SPS*, achieving lower final error and faster early progress. The IMA experiments (right panel) paint the same picture: $\text{IMA-SPS}_{safe}$ is superior to IMA-SPS, and the momentum coefficient $\lambda_t = 9$ is clearly preferable to $\lambda_t = t$. These results confirm that the safeguard delivers practical gains in addition to its theoretical advantages.

### 4.2. Applications on DNNs

**Comparison with other optimizers.** We assess the safeguarded step sizes on image-classification benchmarks. ResNet-20/32 (He et al., 2016) models are trained on CIFAR-10/100 (Krizhevsky et al., 2009) with standard augmentation (random crop, horizontal flip, channel-wise normalisation (DeVries, 2017)). All runs are executed on NVIDIA RTX 6000 Ada GPUs for 100 epochs. Baselines include tuned SSM, tuned IMA with $\lambda_t = 9$, Adam (Kingma & Ba, 2015), and $\text{SPS}_{max}$ and IMA-SPS. We compare these against the proposed safeguarded rules $\text{SPS}_{safe}$ and IMA-$\text{SPS}_{safe}$. Unless stated otherwise, all deep-learning experiments (Figures 3 and 4 and Figures 5 to 7) use a *fixed* safeguard $M = 1$ and lower bounds $\ell_i^* = 0$, the latter being a valid choice since the cross-entropy training loss is nonnegative. We recommend $M = 1$ as a robust default for deep-learning practitioners (see the sensitivity study in Appendix D). The tuning-free smoothed variant $M_t$ is evaluated separately in Appendix E. For more details and more experiments we refer to the appendix. In Figure 3, we observe that in both SSM-based and IMA-based methods our proposed safeguarded step sizes have superior generalization performance.

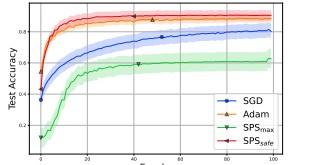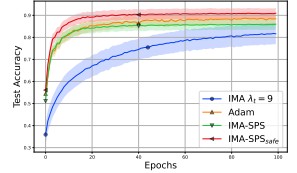

*Figure 3.* Test accuracy of ResNet20 on CIFAR-10. **Left:** SSM-based methods. **Right:** IMA-based methods.

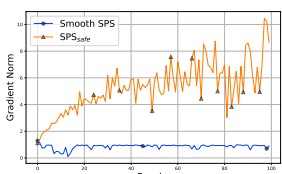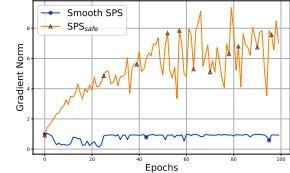

*Figure 4.* Gradient Norms during training of ResNet20. **Left:** Trained on CIFAR-10. **Right:** Trained on CIFAR-100.

**Comparison of Gradient Norms.** In Figure 4, we track the subgradient magnitude $\|g_i^t\|$ at the end of each epoch when training with 2 versus $\text{SPS}_{safe}$ for ResNet-20 in CIFAR-10/100. Empirically, 2 drives (sub)gradients to very small values, whereas under $\text{SPS}_{safe}$ we observe noticeably larger norms. While $\text{SPS}_{safe}$ prevents division by vanishing subgradients in its denominator *by construction*, here we additionally observe *empirically* that the subgradient norms themselves remain larger than under 2, instead of collapsing towards zero.

The gradient norms increase during the training run. However, this increase does not lead to a loss in generalization performance. This observation has recently been studied in Defazio (2025) where the author provides a theoretical explanation for this phenomenon. Here we provide evidence that this phenomenon appears in the safeguarded version of SPS.

## 5. Conclusion and Future Directions

In this work, we introduced *safeguarded* Polyak step sizes and established an $\mathcal{O}(T^{-1/2})$ convergence rate for stochastic convex Lipschitz objectives, without assuming interpolation or requiring knowledge of the component-wise optimal values $f_i(x^*)$. To the best of our knowledge, this provides one of the first Polyak-type rules with such guarantees in this general setting. We further extended this idea to IMA, proving convergence guarantees for both the Cesàro average and the *last* iterate.

We emphasize that our theory is developed for convex Lipschitz objectives, while our deep-learning experiments involve highly non-convex models. Nevertheless, this connection is consistent with the recent findings of Schaipp et al. (2025), who show that convergence bounds for possibly nonsmooth convex Lipschitz objectives can closely track the empirical loss curves observed in large non-convex models.

Several promising directions remain open. On the theoretical side, extending the analysis to structured non-convex settings, such as weakly convex or Polyak-Lojasiewicz objectives (Karimi et al., 2016), as well as to the $(L_0, L_1)$-smooth regime (Zhang et al., 2020), would help bridge the gap between the current convex theory and modern non-convex training. On the algorithmic side, combining $\text{SPS}_{safe}$ with recent training paradigms, particularly schedule-free optimization (Defazio et al., 2024), is a natural next step. Finally, on the applications side, evaluating safeguarded Polyak step sizes in distributed and federated optimization, such as Local SGD (Koloskova et al., 2020), and in the training of LLMs is especially compelling, as these settings combine large-scale stochasticity, sensitivity to learning-rate schedules, and regimes in which automatic step-size adaptation may offer substantial practical benefits.

## Impact Statement

This paper presents work whose goal is to advance the field of Machine Learning. There are many potential societal consequences of our work, none of which we feel must be specifically highlighted here.

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

# Supplementary Material

The Supplementary Material is organized as follows: In Appendix A, we have more details on adaptive Polyak-type step sizes. Appendix B, presents the proofs of the theoretical guarantees from the main paper. In Appendix C, we provide additional experiments. In Appendix D we present more sensitivity analysis plots for DL and in Appendix E we present a practical, smoothed way of calculating the safeguard $M$, together with convergence guarantees for time-varying safeguards.

## A. Further Related Work

Recent advances in Polyak-type step sizes have extended their theory and applicability in both smooth and non-smooth settings. For example, Horváth et al. (2022) introduced StoPS and GraDS, stochastic adaptive step size schemes based on Polyak's rule (and gradient "diversity"), and proved that these methods achieve near-deterministic convergence rates for strongly convex problems. Gower et al. (2022) recast SPS in an online learning framework, showing that SPS and its variants can be viewed as passive-aggressive algorithms, and proposed a single slack variable technique to stabilize the step size in non-interpolated models. This slack-based SPS variant comes with convergence guarantees and improved robustness in practice. To handle regularization, Schaipp et al. (2023) developed ProxSPS, a proximal Polyak step size that only requires a lower bound on the loss (not the entire objective), making it easier to tune and more stable under weight decay. Empirically, ProxSPS performs on par with well-tuned optimizers like AdamW, while requiring minimal tuning. In the non-smooth regime, Zamani & Glineur (2024) established a exact last-iterate convergence rate of the subgradient method with a Polyak step size, showing an $O(N^{-1/4})$ rate that is tight, and proposed an adaptive Polyak step size variant that attains the optimal $O(N^{-1/2})$ rate for convex problems.

Beyond the works discussed above, several recent papers further extend stochastic Polyak step sizes. Liu & Huang (2025) combine the Polyak rule with both variance reduction and momentum, obtaining a stochastic Polyak stepsize for SGD with variance reduction and momentum that improves robustness on ill-conditioned problems. Building on SPS in structured settings, stochastic Polyak steps have been integrated into sparse recovery and high-dimensional statistics: a stochastic IHT method with Polyak step sizes has been proposed for sparse signal recovery, see Li et al. (2024), while the "Sparse Polyak" rule, Qiao & Maros (2024), tailors an adaptive Polyak-type step to high-dimensional M-estimation with sparsity constraints. Finally, preconditioned Polyak step sizes have been studied in the context of SGD, where the SPS denominator is modified by diagonal or adaptive preconditioners, bridging Polyak-type updates with AdaGrad/Adam-style geometry, see Abdukhakimov et al. (2024).

Complementary lines of work exploit loss-landscape structure and Gauss–Newton-type constructions to design related adaptive stepsizes. In Islamov et al. (2024), the authors analyze neural-network objectives beyond the heavily over-parameterized regime and identify geometric conditions (in the spirit of Polyak–Lojasiewicz-type properties) under which gradient methods enjoy fast convergence, thereby providing a landscape-based justification for adaptive step rules closely related to Polyak's idea. Shi et al. (2023) provides an adaptive SARAH-type algorithm that explores and adapts to the local geometry. In a different direction, Orvieto & Xiao (2024) proposes an SGD variant whose effective learning rate is determined by a non-negative Gauss–Newton scaling of the loss; the resulting method exhibits an automatic warm-up and decay behaviour and is analyzed for smooth convex and non-convex problems, with explicit comparisons to Polyak-style and SPS-type stepsizes.

# B. Proofs

In this section, we present the proofs of the main theoretical results presented in the main paper, i.e. Proposition 2.1 and the convergence guarantees of SPS$_{safe}$ and IMA-SPS$_{safe}$. We restate the main theorems here for completeness.

## B.1. Proof of Proposition 2.1

**Proposition B.1.** SSM with SPS$_{safe}$ and $M = c^2$ is algebraically equivalent to Clipped SSM with the *adaptive* step size $\tilde{\gamma}_t = \frac{f_i(x^t) - \ell_i^*}{c \max\{c, \|g_i^t\|\}}$.

*Proof.* We have

$$
\begin{aligned}
x^{t+1} = x^t - \gamma_t g_i^t &= x^t - \frac{f_i(x^t) - \ell_i^*}{\max\{c^2, \|g_i^t\|^2\}} g_i^t \\
&= x^t - \frac{f_i(x^t) - \ell_i^*}{\min\left\{1, \frac{c}{\|g_i^t\|}\right\} \max\{c^2, \|g_i^t\|^2\}} \min\left\{1, \frac{c}{\|g_t\|}\right\} g_i^t \\
&= x^t - \frac{f_i(x^t) - \ell_i^*}{\min\left\{1, \frac{c}{\|g_i^t\|}\right\} \max\{c^2, \|g_i^t\|^2\}} \mathrm{clip}_c\left(g_i^t\right) \\
&= x^t - \frac{f_i(x^t) - \ell_i^*}{c \max\{c, \|g_i^t\|\}} \mathrm{clip}_c\left(g_i^t\right),
\end{aligned}
$$

where the last equality follows by discriminating cases:

- If $\|g_i^t\| \leq c$, then $\max\{c, \|g_i^t\|\} = c$, so:

$$
\min\left\{1, \frac{c}{\|g_i^t\|}\right\} \max\{c^2, \|g_i^t\|^2\} = 1 \cdot c^2 = c^2 \quad \text{and}
$$
$$
c \max\{c, \|g_i^t\|\} = c \cdot c = c^2.
$$

- If $\|g_i^t\| > c$, then $\max\{c, \|g_i^t\|\} = \|g_i^t\|$, so:

$$
\min\left\{1, \frac{c}{\|g_i^t\|}\right\} \max\{c^2, \|g_i^t\|^2\} = \frac{c}{\|g_i^t\|} \cdot \|g_i^t\|^2 = c\|g_i^t\| \quad \text{and}
$$
$$
c \max\{c, \|g_i^t\|\} = c\|g_i^t\|.
$$

This completes the proof. □

## B.2. Proof of Theorem 3.1

**Theorem B.2.** Consider the iterates of SSM with the step size (SPS$_{safe}$). Then

$$
\mathbb{E}[f(\overline{x}^T) - f(x^*)] \leq \frac{\sqrt{\max\{G^2, M\}}\|x^0 - x^*\|}{\sqrt{T}} + \sqrt{\frac{\max\{G^2, M\}}{M}}\sigma^2,
$$

where $\overline{x}^T = \frac{1}{T} \sum_{t=0}^{T-1} x^t$.

*Proof of Theorem 3.1.* We have

$$
\begin{aligned}
\|x^{t+1} - x^*\|^2 - \|x^t - x^*\|^2 &\stackrel{SSM}{=} -2\gamma_t \langle g_i^t, x^t - x^* \rangle + \gamma_t^2 \|g_i^t\|^2 \\
&\stackrel{\text{convexity}}{\leq} -2\gamma_t [f_i(x^t) - f_i(x^*)] + \gamma_t^2 \|g_i^t\|^2 \\
&\stackrel{SPS_{safe}}{=} -\frac{2[f_i(x^t) - \ell_i^*][f_i(x^t) - f_i(x^*)]}{\max\{\|g_i^t\|^2, M\}} + \frac{[f_i(x^t) - \ell_i^*]^2}{(\max\{\|g_i^t\|^2, M\})^2}\|g_i^t\|^2
\end{aligned}
$$

$$
\begin{aligned}
&= -\frac{2[f_i(x^t) - \ell_i^*][f_i(x^t) - f_i(x^*)]}{\max\{\|g_i^t\|^2, M\}} + \frac{[f_i(x^t) - \ell_i^*]^2}{\max\{\|g_i^t\|^2, M\}} \frac{\|g_i^t\|^2}{\max\{\|g_i^t\|^2, M\}} \\
&\overset{(\star)}{\leq} -\frac{2[f_i(x^t) - \ell_i^*][f_i(x^t) - f_i(x^*)]}{\max\{\|g_i^t\|^2, M\}} + \frac{[f_i(x^t) - \ell_i^*]^2}{\max\{\|g_i^t\|^2, M\}} \\
&= \frac{-2[f_i(x^t) - \ell_i^*][f_i(x^t) - f_i(x^*)] + [f_i(x^t) - \ell_i^*]^2}{\max\{\|g_i^t\|^2, M\}} \\
&\overset{(\star\star)}{=} \frac{(f_i(x^*) - \ell_i^*)^2 - (f_i(x^t) - f_i(x^*))^2}{\max\{\|g_i^t\|^2, M\}} \\
&= -\frac{(f_i(x^t) - f_i(x^*))^2}{\max\{\|g_i^t\|^2, M\}} + \frac{(f_i(x^*) - \ell_i^*)^2}{\max\{\|g_i^t\|^2, M\}} \\
&\overset{(\star\star\star)}{\leq} -\frac{(f_i(x^t) - f_i(x^*))^2}{\max\{G^2, M\}} + \frac{(f_i(x^*) - \ell_i^*)^2}{M}.
\end{aligned} \tag{5}
$$

The inequality $(\star)$ follows from the fact that $\max\{\|g_i^t\|^2, M\} \geq \|g_i^t\|^2$, the equality $(\star\star)$ follows from the identity $-2xy + y^2 = (x-y)^2 - x^2$ with $x = f_i(x^t) - f_i(x^*)$ and $y = f_i(x^t) - \ell_i^*$. Finally the inequality (5) follows from $\max\{\|g_i^t\|^2, M\} \geq M$ and $\|g_i^t\| \leq G$. Taking expectation conditional on $x^t$ at $(*)$ we get

$$
\begin{aligned}
\mathbb{E}[\|x^{t+1} - x^*\|^2 | x^t] - \|x^t - x^*\|^2 &\leq -\frac{\mathbb{E}\left([f_i(x^t) - f_i(x^*)]\right)^2}{\max\{G^2, M\}} + \frac{\sigma^4}{M} \\
&\leq -\frac{(\mathbb{E}[f_i(x^t) - f_i(x^*)])^2}{\max\{G^2, M\}} + \frac{\sigma^4}{M} \\
&= -\frac{[f(x^t) - f(x^*)]^2}{\max\{G^2, M\}} + \frac{\sigma^4}{M},
\end{aligned}
$$

where the second inequality follows by Jensen's inequality. Taking expectation again and using the tower property we have

$$
\mathbb{E}\|x^{t+1} - x^*\|^2 \leq \mathbb{E}\|x^t - x^*\|^2 - \frac{\mathbb{E}[f(x^t) - f(x^*)]^2}{\max\{G^2, M\}} + \frac{\sigma^4}{M},
$$

or equivalently, after rearranging:

$$
\mathbb{E}[f(x^t) - f(x^*)]^2 \leq \max\{G^2, M\}\mathbb{E}\|x^t - x^*\|^2 - \max\{G^2, M\}\mathbb{E}\|x^{t+1} - x^*\|^2 + \frac{\max\{G^2, M\}\sigma^4}{M}. \tag{6}
$$

Now summing up (6) for $t = 0, \dots, T-1$ and telescoping we have

$$
\begin{aligned}
\sum_{t=0}^{T-1} \mathbb{E}[f(x^t) - f(x^*)]^2 &\leq \sum_{t=0}^{T-1} \left[ \max\{G^2, M\}\mathbb{E}\|x^t - x^*\|^2 - \max\{G^2, M\}\mathbb{E}\|x^{t+1} - x^*\|^2 + \frac{\max\{G^2, M\}\sigma^4}{M} \right] \\
&= \max\{G^2, M\}\|x^0 - x^*\|^2 - \max\{G^2, M\}\mathbb{E}\|x^T - x^*\|^2 + T\frac{\max\{G^2, M\}\sigma^4}{M} \\
&\leq \max\{G^2, M\}\|x^0 - x^*\|^2 + T\frac{\max\{G^2, M\}\sigma^4}{M}.
\end{aligned} \tag{7}
$$

Now by Jensen's inequality (twice) we get

$$
\begin{aligned}
\mathbb{E}[f(\overline{x}^T) - f(x^*)] &\overset{\text{Jensen}}{\leq} \frac{1}{T}\sum_{t=0}^{T-1} \mathbb{E}[f(x^t) - f(x^*)] \\
&\overset{\text{Jensen}}{\leq} \sqrt{\frac{1}{T}\sum_{t=0}^{T-1} \mathbb{E}[f(x^t) - f(x^*)]^2} \\
&\overset{(7)}{\leq} \sqrt{\frac{\max\{G^2, M\}\|x^0 - x^*\|^2}{T} + \frac{\max\{G^2, M\}\sigma^4}{M}}
\end{aligned}
$$

$$\leq \frac{\sqrt{\max\{G^2, M\}}\|x^0 - x^*\|^2}{\sqrt{T}} + \sqrt{\frac{\max\{G^2, M\}}{M}}\sigma^2,$$

because $\sqrt{x+y} \leq \sqrt{x} + \sqrt{y}$. This completes the proof. $\qquad\square$

## B.3. Proofs of Theorems 3.5 and 3.6

### B.3.1. PRELIMINARY LEMMAS

Here we provide the two auxiliary lemmas we will use in the following proofs.

**Lemma B.3** ((Gower et al., 2025): Lem. B.3). For any random variable $X$ and positive-valued random variable $Y$, it holds

$$\mathbb{E}\left[\frac{(X)_+^2}{Y}\right] \geq \frac{(\mathbb{E}X)_+^2}{\mathbb{E}Y}.$$

**Lemma B.4** ((Gower et al., 2025): Lem. C.3). For any $x^{t-1}, x^t, x^* \in \mathbb{R}^d$, any $\lambda_t \geq 0$, and any subgradient $g^t \in \partial f(x^t)$, it holds

$$f(x^t) - f(x^*) + \lambda_t \langle g^t, x^t - x^{t-1} \rangle$$
$$= (1 + \lambda_t)[f(x^t) - f(x^*)] - \lambda_t[f(x^{t-1}) - f(x^*)] + \lambda_t B_f(x^{t-1}, x^t),$$

where $B_f(x^{t-1}, x^t) = f(x^{t-1}) - f(x^t) - \langle g^t, x^{t-1} - x^t \rangle$ is the Bregman divergence with respect to $g^t$.

### B.3.2. PROOF OF THEOREM 3.5

**Theorem B.5.** Consider the iterates of IMA with the step size (IMA-SPS$_{safe}$) and let $(\lambda_t)_{t>0}$ be a non-increasing sequence of nonnegative reals. Then

$$\mathbb{E}[f(\overline{x}^T) - f(x^*)] + \sum_{t=0}^{T-1} \frac{\lambda_t}{T}\mathbb{E}[B_f(x^{t-1}, x^t)] \leq \frac{\sqrt{\max\{G^2, M\}}\|x^0 - x^*\|}{\sqrt{T}} + \sqrt{\frac{\max\{G^2, M\}}{M}}\sigma^2,$$

where $\overline{x}^T = \frac{1}{T}\sum_{t=0}^{T-1} x^t$ and $B_f$ is the Bregman divergence defined in Remark 3.4.

*Proof.* We have

$$\|z^{t+1} - x^*\|^2 - \|z^t - x^*\|^2 \overset{(IMA)}{=} -2\eta_t \langle g_i^t, z^t - z^* \rangle + \eta_t^2 \|g_i^t\|^2$$
$$= -2\eta_t \langle g_i^t, x^t - x^* \rangle - 2\eta_t \lambda_t \langle g_i^t, x^t - x^{t-1} \rangle + \eta_t^2 \|g_i^t\|^2$$
$$\overset{\text{convexity}}{\leq} -2\eta_t[f_i(x^t) - f_i(x^*) + \lambda_t \langle g_i^t, x^t - x^{t-1} \rangle] + \eta_t^2 \|g_i^t\|^2$$
$$= -\frac{2[f_i(x^t) - \ell_i^* + \lambda_t \langle g_i^t, x^t - x^{t-1} \rangle]_+ [f_i(x^t) - f_i(x^*) + \lambda_t \langle g_i^t, x^t - x^{t-1} \rangle]}{\max\{\|g_i^t\|^2, M\}}$$
$$+ \frac{[f_i(x^t) - \ell_i^* + \lambda_t \langle g_i^t, x^t - x^{t-1} \rangle]_+^2}{(\max\{\|g_i^t\|^2, M\})^2}\|g_i^t\|^2.$$

Now for easier notation we set $q = f_i(x^t) + \lambda_t \langle g_i^t, x^t - x^{t-1} \rangle$, thus we continue:

$$\|z^{t+1} - x^*\|^2 - \|z^t - x^*\|^2$$
$$= \frac{-2(q - \ell_i^*)_+ \cdot (q - f_i(x^*))}{\max\{\|g_i^t\|^2, M\}} + \frac{(q - \ell_i^*)_+^2}{\max\{\|g_i^t\|^2, M\}} \cdot \frac{\|g_i^t\|^2}{\max\{\|g_i^t\|^2, M\}}$$
$$\leq \frac{-2(q - \ell_i^*)_+ \cdot (q - f_i(x^*))}{\max\{\|g_i^t\|^2, M\}} + \frac{(q - \ell_i^*)_+^2}{\max\{\|g_i^t\|^2, M\}}$$
$$= \frac{-2(q - \ell_i^*)_+ \cdot (q - f_i(x^*)) + (q - \ell_i^*)_+^2}{\max\{\|g_i^t\|^2, M\}}$$

$$\overset{(\star)}{\leq} \frac{(f_i(x^*) - \ell_i^*)^2 - (q - f_i(x^*))_+^2}{\max\{\|g_i^t\|^2, M\}}$$

$$= -\frac{(f_i(x^t) - f_i(x^*) + \lambda_t \langle g_i^t, x^t - x^{t-1}\rangle)_+^2}{\max\{\|g_i^t\|^2, M\}} + \frac{[f_i(x^*) - \ell_i^*]^2}{\max\{\|g_i^t\|^2, M\}}$$

$$\overset{(\star\star)}{\leq} -\frac{[f_i(x^t) - f_i(x^*) + \lambda_t \langle g_i^t, x^t - x^{t-1}\rangle]_+^2}{\max\{G^2, M\}} + \frac{[f_i(x^*) - \ell_i^*]^2}{M}.$$

Let's explain inequality $(\star)$, which is:

$$-2(q - \ell_i^*)_+ \cdot (q - f_i(x^*)) + (q - \ell_i^*)_+^2 \leq (f_i(x^*) - \ell_i^*)^2 - (q - f_i(x^*))_+^2 \qquad (\star)$$

Note that $\ell_i^* \leq f_i(x^*)$ so $q - \ell_i^* \geq q - f_i(x^*)$. Hence if $q - \ell_i^* \leq 0$ inequality $(\star)$ reduces to the obvious $0 \leq [f_i(x^t) - \ell_i^*]^2$. Now assume that $q - \ell_i^* > 0$. Then

$$-2(q - f_i^*)_+ \cdot (q - f_i(x^*)) + (q - f_i^*)_+^2 = -2(q - f_i^*)(q - f_i(x^*)) + (q - f_i^*)^2$$

$$= (q - f_i^* - (q - f_i(x^*)))^2 - (q - f_i(x^*))^2$$

$$= (f_i(x^*) - f_i^*)^2 - (q - f_i(x^*))^2$$

$$\leq (f_i(x^*) - f_i^*)^2 - (q - f_i(x^*))_+^2,$$

as wanted. Now, inequality $(\star\star)$ follows from $\max\{\|g_i^t\|^2, M\} \geq M$ and $\max\{\|g_i^t\|^2, M\} \leq \max\{G^2, M\}$ because $f_i$ is $G$-Lipschitz.

Now taking expectation, so that the cross term becomes $\mathbb{E}_i[\lambda_t \langle g_i^t, x^t - x^{t-1}\rangle \mid x^t] = \lambda_t \langle g^t, x^t - x^{t-1}\rangle$ with $g^t := \mathbb{E}_i[g_i^t \mid x^t] \in \partial f(x^t)$ as in Remark 3.4, and using Lemmas B.3 and B.4, we get

$$\mathbb{E}\|z^{t+1} - x^*\|^2 \leq \mathbb{E}\|z^t - x^*\|^2 - \frac{\mathbb{E}[f(x^t) - f(x^*) + \lambda_t \langle g^t, x^t - x^{t-1}\rangle]_+^2}{\max\{G^2, M\}} + \frac{\sigma^4}{M}$$

$$= \mathbb{E}\|z^t - x^*\|^2 - \frac{\mathbb{E}[(1 + \lambda_t)[f(x^t) - f(x^*)] - \lambda_t[f(x^{t-1}) - f(x^*)] + \lambda_t B_f(x^{t-1}, x^t)]_+^2}{\max\{G^2, M\}} + \frac{\sigma^4}{M},$$

so

$$\mathbb{E}[(1 + \lambda_t)[f(x^t) - f(x^*)] - \lambda_t[f(x^{t-1}) - f(x^*)] + \lambda_t B_f(x^{t-1}, x^t)]_+^2$$

$$\leq \max\{G^2, M\}\mathbb{E}\|z^t - x^*\|^2 - \max\{G^2, M\}\mathbb{E}\|z^{t+1} - x^*\|^2 + \frac{\max\{G^2, M\}}{M}\sigma^4.$$

Now let $\Delta_t = (1 + \lambda_t)[f(x^t) - f(x^*)] - \lambda_t[f(x^{t-1}) - f(x^*)] + \lambda_t B_f(x^{t-1}, x^t)$, sum for $t = 0, \ldots, T - 1$ and use Jensen to get

$$\frac{\max\{G^2, M\}\|x^0 - x^*\|^2}{T} + \frac{\max\{G^2, M\}}{M}\sigma^4$$

$$\geq \frac{1}{T}\sum_{t=0}^{T-1} \mathbb{E}[\Delta_t]_+^2$$

$$\geq \left(\frac{1}{T}\sum_{t=0}^{T-1} \mathbb{E}[\Delta_t]\right)_+^2,$$

which means that

$$\left(\frac{1}{T}\sum_{t=0}^{T-1} \mathbb{E}[\Delta_t]\right)_+ \leq \sqrt{\frac{G^2\|x^0 - x^*\|^2}{T} + \frac{\max\{G^2, M\}}{M}\sigma^4}$$

$$\leq \frac{\sqrt{\max\{G^2, M\}}\|x^0 - x^*\|}{\sqrt{T}} + \sqrt{\frac{\max\{G^2, M\}}{M}}\sigma^2. \tag{8}$$

Now

$$\sum_{t=0}^{T-1} \mathbb{E}[\Delta_t] = \sum_{t=0}^{T-1}(1 + \lambda_t)\mathbb{E}[f(x^t) - f(x^*)] - \lambda_t \mathbb{E}[f(x^{t-1}) - f(x^*)] + \lambda_t \mathbb{E}[B_f(x^{t-1}, x^t)]$$

$$= \sum_{t=0}^{T-1} \lambda_t \mathbb{E}[B_f(x^{t-1}, x^t)] + \sum_{t=0}^{T-1} \mathbb{E}[f(x^t) - f(x^*)] + \sum_{t=0}^{T-2}(\lambda_t - \lambda_{t+1})\mathbb{E}[f(x^t) - f(x^*)]$$

$$+ \lambda_{T-1} \mathbb{E}[f(x^{T-1}) - f(x^*)]. \tag{9}$$

Since $(\lambda_t)_{t>0}$ is non-increasing we get

$$\sum_{t=0}^{T-1} \lambda_t \mathbb{E}[B_f(x^{t-1}, x^t)] + \sum_{t=0}^{T-1} \mathbb{E}[f(x^t) - f(x^*)] + \sum_{t=0}^{T-2}(\lambda_t - \lambda_{t+1})\mathbb{E}[f(x^t) - f(x^*)]$$

$$+ \lambda_{T-1} \mathbb{E}[f(x^{T-1}) - f(x^*)]$$

$$\geq \sum_{t=0}^{T-1} \lambda_t \mathbb{E}[B_f(x^{t-1}, x^t)] + \sum_{t=0}^{T-1} \mathbb{E}[f(x^t) - f(x^*)]$$

$$\geq \sum_{t=0}^{T-1} \lambda_t \mathbb{E}[B_f(x^{t-1}, x^t)] + T \cdot \mathbb{E}[f(\overline{x}^T) - f(x^*)],$$

so, dividing by $T$, we get

$$\mathbb{E}[f(\overline{x}^T) - f(x^*)] + \sum_{t=0}^{T-1} \frac{\lambda_t}{T} \mathbb{E}[B_f(x^{t-1}, x^t)] \leq \frac{\sqrt{\max\{G^2, M\}}\|x^0 - x^*\|}{\sqrt{T}} + \sqrt{\frac{\max\{G^2, M\}}{M}}\sigma^2.$$

This completes the proof. $\qquad\square$

### B.3.3. PROOF OF THEOREM 3.6

**Theorem B.6.** Consider the iterates of IMA with the step size (IMA-SPS$_{safe}$) and let $\lambda_t = t$. Then

$$\mathbb{E}[f(x^{T-1}) - f(x^*)] + \frac{1}{T}\sum_{t=0}^{T-1} t\mathbb{E}[B_f(x^{t-1}, x^t)] \leq \frac{\sqrt{\max\{G^2, M\}}\|x^0 - x^*\|}{\sqrt{T}} + \sqrt{\frac{\max\{G^2, M\}}{M}}\sigma^2.$$

*Proof.* The proof is very similar to the proof of Theorem 3.5. Indeed by Equation (8) we have

$$\left(\frac{1}{T}\sum_{t=0}^{T-1} \mathbb{E}[\Delta_t]\right)_+ \leq \frac{\sqrt{\max\{G^2, M\}}\|x^0 - x^*\|}{\sqrt{T}} + \sqrt{\frac{\max\{G^2, M\}}{M}}\sigma^2,$$

and by Equation (9) we get

$$\sum_{t=0}^{T-1} \mathbb{E}[\Delta_t] = \sum_{t=0}^{T-1} \lambda_t \mathbb{E}[B_f(x^{t-1}, x^t)] + \sum_{t=0}^{T-1} \mathbb{E}[f(x^t) - f(x^*)] + \sum_{t=0}^{T-2}(\lambda_t - \lambda_{t+1})\mathbb{E}[f(x^t) - f(x^*)]$$

$$+ \lambda_{T-1} \mathbb{E}[f(x^{T-1}) - f(x^*)].$$

Now since $\lambda_t = t$ we have

$$\sum_{t=0}^{T-1} \lambda_t \mathbb{E}[B_f(x^{t-1}, x^t)] + \sum_{t=0}^{T-1} \mathbb{E}[f(x^t) - f(x^*)] + \sum_{t=0}^{T-2}(\lambda_t - \lambda_{t+1})\mathbb{E}[f(x^t) - f(x^*)]$$

$$+ \lambda_{T-1} \mathbb{E}[f(x^{T-1}) - f(x^*)]$$

$$= \sum_{t=0}^{T-1} t\mathbb{E}[B_f(x^{t-1}, x^t)] + T \cdot \mathbb{E}[f(x^{T-1}) - f(x^*)] \geq 0,$$

so we get

$$\mathbb{E}[f(x^{T-1}) - f(x^*)] + \sum_{t=0}^{T-1} \frac{t}{T} \mathbb{E}[B_f(x^{t-1}, x^t)] \leq \frac{\sqrt{\max\{G^2, M\}} \|x^0 - x^*\|}{\sqrt{T}} + \sqrt{\frac{\max\{G^2, M\}}{M}} \sigma^2.$$

This completes the proof. □

# C. Further Numerical Experiments

## C.1. More deep learning experiments and parameter settings

In this section, we list the parameters, architectures and hardware that we used for the deep learning experiments. The information is collected in Table 2. We also include some extra experiments (ResNet20 in CIFAR-100 and ResNet32 in CIFAR-10/100) in Figures 5 to 7.

| Hyper-parameter | Value |
|---|---|
| Datasets | CIFAR-10/100 (Krizhevsky et al., 2009) |
| Architecture | ResNet 20/32 (He et al., 2016) |
| GPUs | 1x Nvidia RTX 6000 Ada Generation |
| Batch-size | 128 |
| Epochs | 100 |
| Weight Decay | 0.0 |

*Table 2.* Experimental details

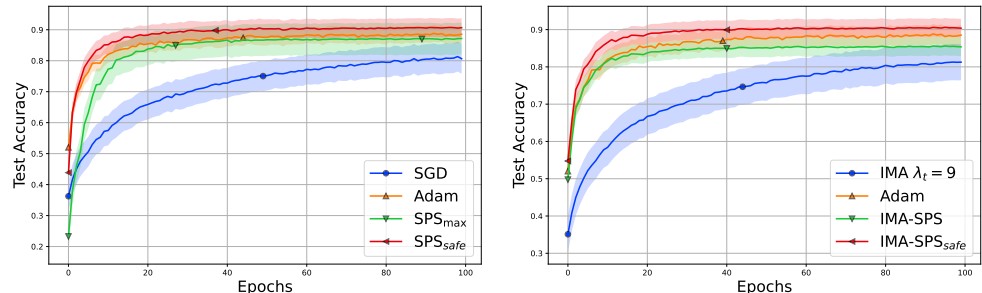

*Figure 5.* Test accuracy of ResNet20 on CIFAR-100. **Left:** SSM-based methods. **Right:** IMA-based methods.

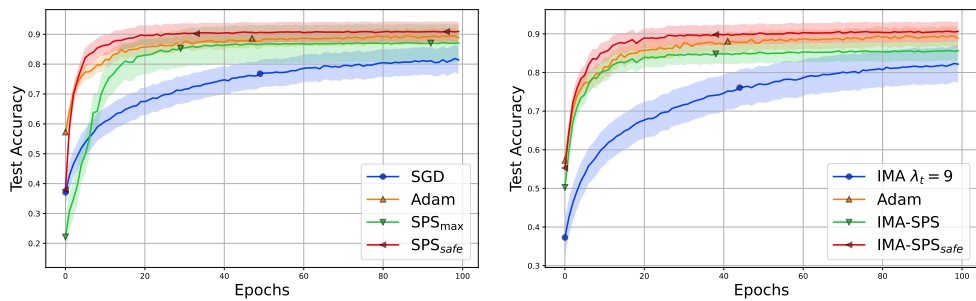

*Figure 6.* Test accuracy of ResNet32 on CIFAR-10. **Left:** SSM-based methods. **Right:** IMA-based methods.

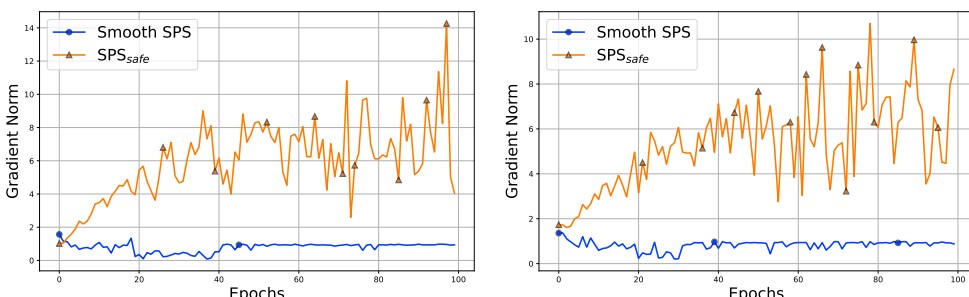

*Figure 7.* Test accuracy of ResNet32 on CIFAR-100. **Left:** SSM-based methods. **Right:** IMA-based methods.

### C.1.1. COMPARISON OF THE GRADIENT NORM

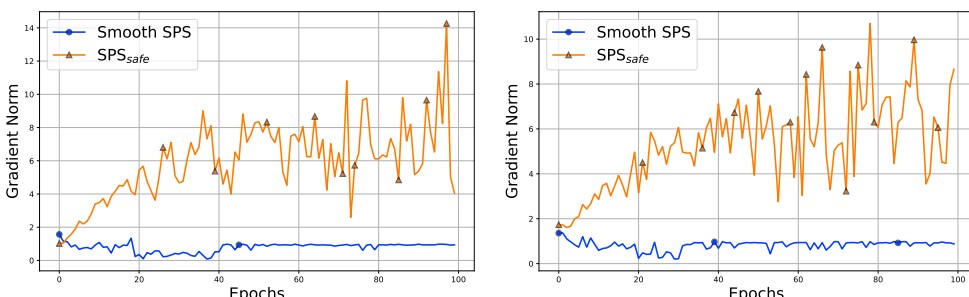

*Figure 8.* Gradient Norms during training of ResNet20. **Left:** Trained on CIFAR-10. **Right:** Trained on CIFAR-100.

# D. Extra Sensitivity Analysis

In this appendix we complement the main sensitivity study for the safeguard $M$ by providing additional experiments on both convex and deep-learning benchmarks. We systematically vary $M$ over a wide range and report the generalization performance vs the value of $M$ for SSM and IMA variants. These plots illustrate that choosing $M = 1.0$ works well in practice.

## D.1. Convex

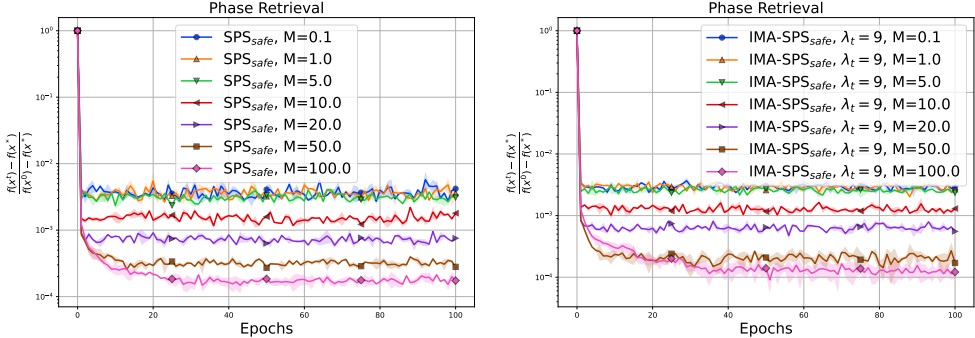

*Figure 9.* Sensitivity Analysis for various safeguards $M$ for Phase Retrieval. **Left:** SSM. **Right:** IMA

## D.2. Deep Learning

### D.2.1. SSM

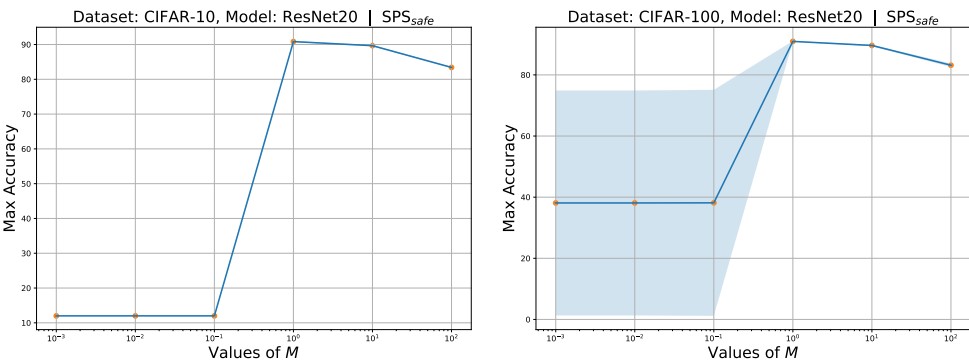

*Figure 10.* Sensitivity Analysis for various safeguards $M$ for ResNet20. **Left:** Trained on CIFAR-10. **Right:** Trained on CIFAR-100.

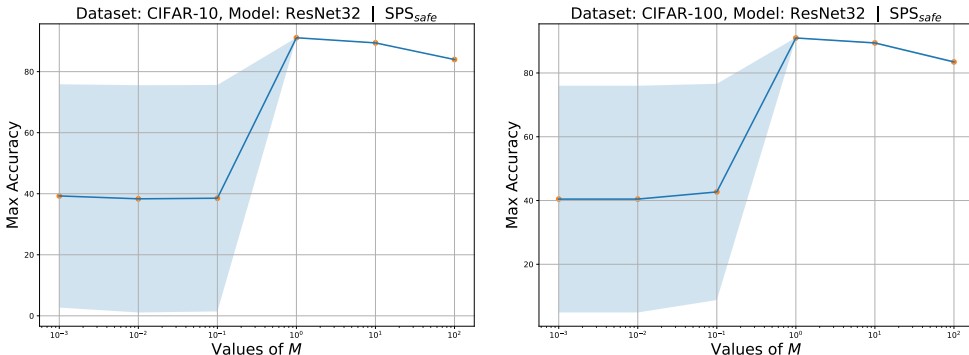

*Figure 11.* Sensitivity Analysis for various safeguards $M$ for ResNet32. **Left:** Trained on CIFAR-10. **Right:** Trained on CIFAR-100.

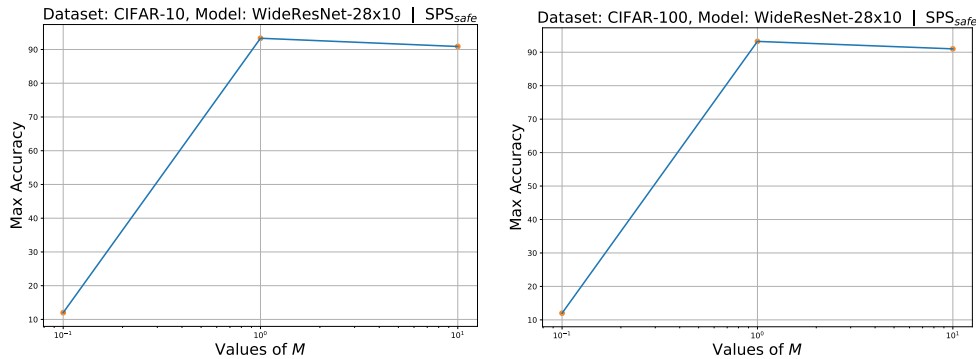

*Figure 12.* Sensitivity Analysis for various safeguards $M$ for WideResNet 28x10. **Left:** Trained on CIFAR-10. **Right:** Trained on CIFAR-100.

### D.2.2. IMA

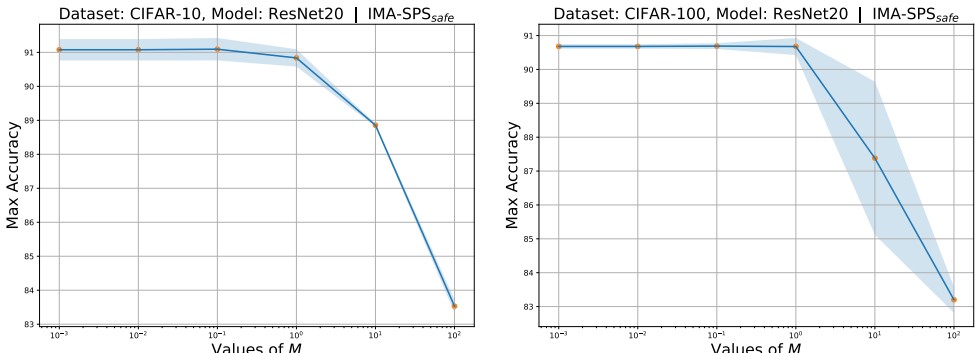

*Figure 13.* Sensitivity Analysis for various safeguards $M$ for ResNet20. **Left:** Trained on CIFAR-10. **Right:** Trained on CIFAR-100.

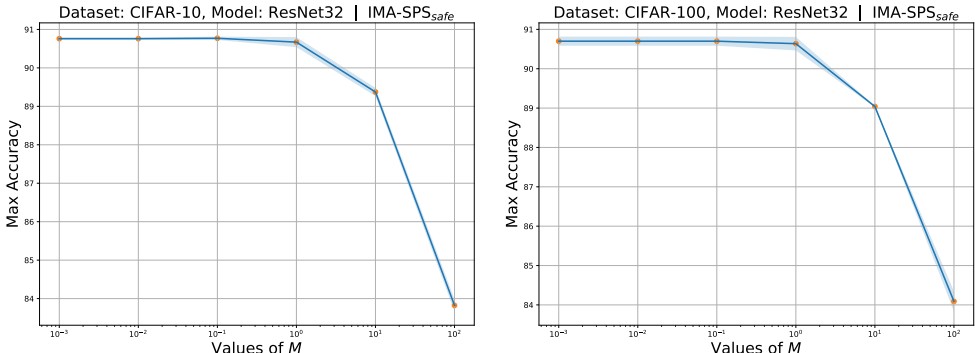

*Figure 14.* Sensitivity Analysis for various safeguards $M$ for ResNet32. **Left:** Trained on CIFAR-10. **Right:** Trained on CIFAR-100.

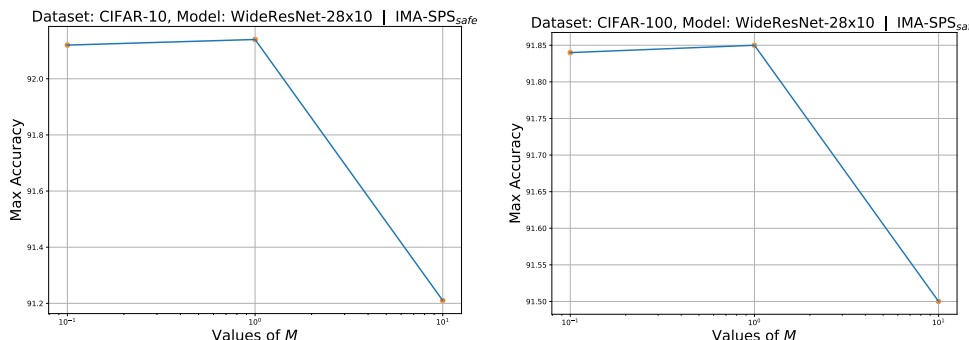

*Figure 15.* Sensitivity Analysis for various safeguards $M$ for WideResNet 28x10. **Left:** Trained on CIFAR-10. **Right:** Trained on CIFAR-100.

# E. Smoothing trick for $M$

Motivated by the sensitivity analyses above, in this appendix, we investigate a simple smoothing strategy that replaces the fixed safeguard $M$ with an exponential moving average $M_t$ of past squared gradients. This adaptive rule is designed to reduce manual tuning while preserving the stabilizing effect of the safeguard. We present the corresponding update, provide a practical recommendation for the smoothing parameter $\beta$, and compare the resulting $M_t$ against the best-tuned fixed $M$ on CIFAR-10/100 and several architectures.

The SPS$_{safe}$ takes the following form:

$$\gamma_t = \frac{f_i(x^t) - \ell_i^*}{\max\{\|g_i^t\|^2, M_t\}}$$
$$M_t = \beta M_{t-1} + (1 - \beta)\|g_i^t\|^2,$$

with $M_0 = \|g_i^0\|^2$. For a good practical performance we recommend $\beta = 0.9$.

*Table 3.* Comparison of test accuracy of tuned $M$ vs Smooth $M_t$ for various model on CIFAR10.

| Model | Best $M$ | Smooth $M_t$ ($\beta = 0.9$) |
|---|---|---|
| ResNet20 | $90.84_{\pm0.17}$ | $\mathbf{90.97_{\pm0.14}}$ |
| ResNet32 | $90.80_{\pm0.04}$ | $\mathbf{90.94_{\pm0.13}}$ |
| WideResNet-28x10 | $\mathbf{93.03}$ | $92.99$ |

*Table 4.* Comparison of test accuracy of tuned $M$ vs Smooth $M_t$ for various model on CIFAR100.

| Model | Best $M$ | Smooth $M_t$ ($\beta = 0.9$) |
|---|---|---|
| ResNet20 | $90.86_{\pm0.12}$ | $\mathbf{90.93_{\pm0.16}}$ |
| ResNet32 | $90.97_{\pm0.11}$ | $\mathbf{91.11_{\pm0.28}}$ |
| WideResNet-28x10 | $\mathbf{93.24}$ | $93.22$ |

## E.1. Convergence guarantees for a time-varying safeguard

The convergence analysis of Theorem 3.1 extends to a *time-varying* safeguard $M_t$ without changing the proof structure, provided $M_t$ stays uniformly bounded away from zero and from above. This endows smoothed safeguards, such as the exponential moving average above, with convergence guarantees, and is a first step towards a theory for adaptive safeguards.

**Theorem E.1** (Time-varying safeguarded SPS). Consider the iterates of SSM with step size

$$\gamma_t = \frac{f_i(x^t) - \ell_i^*}{\max\{\|g_i^t\|^2, M_t\}},$$

where $g_i^t \in \partial f_i(x^t)$, and where $M_t$ may be random and may depend on the current sample. Assume there exist deterministic constants $0 < m \leq M_{\max}$ such that

$$m \leq M_t \leq M_{\max} \qquad \text{almost surely for all } t.$$

Then, for $\overline{x}^T = \frac{1}{T} \sum_{t=0}^{T-1} x^t$,

$$\mathbb{E}[f(\overline{x}^T) - f(x^*)] \leq \frac{\sqrt{\max\{G^2, M_{\max}\}}\|x^0 - x^*\|}{\sqrt{T}} + \sqrt{\frac{\max\{G^2, M_{\max}\}}{m}}\sigma^2.$$

*Proof.* Using the update $x^{t+1} = x^t - \gamma_t g_i^t$, convexity of $f_i$, and the definition of $\gamma_t$, we obtain

$$\|x^{t+1} - x^*\|^2 - \|x^t - x^*\|^2 = -2\gamma_t \langle g_i^t, x^t - x^* \rangle + \gamma_t^2 \|g_i^t\|^2$$
$$\leq -2\gamma_t \big(f_i(x^t) - f_i(x^*)\big) + \gamma_t^2 \|g_i^t\|^2$$

$$= -\frac{2(f_i(x^t) - \ell_i^*)(f_i(x^t) - f_i(x^*))}{\max\{\|g_i^t\|^2, M_t\}} + \frac{(f_i(x^t) - \ell_i^*)^2}{(\max\{\|g_i^t\|^2, M_t\})^2}\|g_i^t\|^2$$

$$\leq -\frac{2(f_i(x^t) - \ell_i^*)(f_i(x^t) - f_i(x^*))}{\max\{\|g_i^t\|^2, M_t\}} + \frac{(f_i(x^t) - \ell_i^*)^2}{\max\{\|g_i^t\|^2, M_t\}},$$

where the last step uses $\|g_i^t\|^2 \leq \max\{\|g_i^t\|^2, M_t\}$. Now use the identity $-2ab + a^2 = (a-b)^2 - b^2$ with $a = f_i(x^t) - \ell_i^*$ and $b = f_i(x^t) - f_i(x^*)$. Since $a - b = f_i(x^*) - \ell_i^*$, the previous inequality becomes

$$\|x^{t+1} - x^*\|^2 - \|x^t - x^*\|^2 \leq \frac{(f_i(x^*) - \ell_i^*)^2 - (f_i(x^t) - f_i(x^*))^2}{\max\{\|g_i^t\|^2, M_t\}}.$$

Because $\max\{\|g_i^t\|^2, M_t\} \geq m$ and $\max\{\|g_i^t\|^2, M_t\} \leq \max\{G^2, M_{\max}\}$ (using $\|g_i^t\| \leq G$ and $M_t \leq M_{\max}$), we further get

$$\|x^{t+1} - x^*\|^2 - \|x^t - x^*\|^2 \leq -\frac{(f_i(x^t) - f_i(x^*))^2}{\max\{G^2, M_{\max}\}} + \frac{(f_i(x^*) - \ell_i^*)^2}{m}.$$

Taking expectation $\mathbb{E}_i$ conditional on $x^t$, and using Jensen's inequality $\mathbb{E}_i[(f_i(x^t) - f_i(x^*))^2 \mid x^t] \geq (f(x^t) - f(x^*))^2$, gives

$$\mathbb{E}_i[\|x^{t+1} - x^*\|^2 \mid x^t] - \|x^t - x^*\|^2 \leq -\frac{(f(x^t) - f(x^*))^2}{\max\{G^2, M_{\max}\}} + \frac{\sigma^4}{m}.$$

Taking total expectation and rearranging yields

$$\mathbb{E}[f(x^t) - f(x^*)]^2 \leq \max\{G^2, M_{\max}\}\, \mathbb{E}\|x^t - x^*\|^2 - \max\{G^2, M_{\max}\}\, \mathbb{E}\|x^{t+1} - x^*\|^2 + \frac{\max\{G^2, M_{\max}\}}{m}\sigma^4.$$

Summing from $t = 0$ to $T - 1$ and telescoping gives

$$\frac{1}{T}\sum_{t=0}^{T-1} \mathbb{E}[f(x^t) - f(x^*)]^2 \leq \frac{\max\{G^2, M_{\max}\}\|x^0 - x^*\|^2}{T} + \frac{\max\{G^2, M_{\max}\}}{m}\sigma^4.$$

Finally, convexity of $f$ and Jensen's inequality imply

$$\mathbb{E}[f(\overline{x}^T) - f(x^*)] \leq \frac{1}{T}\sum_{t=0}^{T-1} \mathbb{E}[f(x^t) - f(x^*)] \leq \sqrt{\frac{1}{T}\sum_{t=0}^{T-1} \mathbb{E}[f(x^t) - f(x^*)]^2}.$$

Combining the last two displays and using $\sqrt{a + b} \leq \sqrt{a} + \sqrt{b}$ proves the result. $\square$

Theorem E.1 shows that the $\mathcal{O}(T^{-1/2})$ rate of Theorem 3.1 is preserved for *any* safeguard sequence $M_t$ that is confined to a fixed interval $[m, M_{\max}]$, with the neighborhood now controlled by the floor $m$ and the ceiling $M_{\max}$. In particular, it applies to smoothed safeguards once a positive floor is enforced.

**Corollary E.2** (Exponential moving average safeguard with a floor). Assume the safeguard is updated according to

$$M_t = \max\left\{m, \beta M_{t-1} + (1 - \beta)\|g_i^t\|^2\right\}, \qquad 0 \leq \beta < 1,$$

with $M_0 \geq m > 0$. Then, almost surely, $m \leq M_t \leq \max\{M_0, G^2\}$ for all $t$, and hence

$$\mathbb{E}[f(\overline{x}^T) - f(x^*)] \leq \frac{\sqrt{\max\{G^2, M_0\}}\|x^0 - x^*\|}{\sqrt{T}} + \sqrt{\frac{\max\{G^2, M_0\}}{m}}\,\sigma^2.$$

*Proof.* The lower bound $M_t \geq m$ is immediate from the definition. For the upper bound, set $U := \max\{M_0, G^2\}$. We prove by induction that $M_t \leq U$ for all $t$. The claim holds at $t = 0$ since $M_0 \leq U$. If $M_{t-1} \leq U$, then using $\|g_i^t\| \leq G$ we have

$$\beta M_{t-1} + (1 - \beta)\|g_i^t\|^2 \leq \beta U + (1 - \beta)G^2 \leq U.$$

Taking the maximum with $m$ does not increase the quantity above $U$, since $m \leq U$. Hence $M_t \leq U$, and the claim follows by induction. Theorem E.1 then applies with $M_{\max} = U$. $\square$

The exponential moving average used in practice at the beginning of this appendix corresponds to $\beta = 0.9$ *without* an explicit floor. The induction in the proof above shows that it is still provably upper bounded by $\max\{M_0, G^2\}$; adding any positive floor $m$ (for instance a small constant, or $m = M_0$) makes Corollary E.2 directly applicable and yields a convergence guarantee for the smoothed safeguard. Establishing exact convergence (rather than convergence to a neighborhood) for a fully adaptive, floor-free safeguard remains an interesting open question.

