# OpenReview forum: "Safeguarded Stochastic Polyak Step Sizes for Non-smooth Optimization: Robust Performance Without Small (Sub)Gradients"
_ICML.cc/2026/Conference — ICML 2026 regular_

### Official Review · Reviewer_thtm · 2026-03-04

**Soundness:** 3
**Presentation:** 4
**Significance:** 2
**Originality:** 2
**Overall Recommendation:** 5
**Confidence:** 4

**Summary:**

The paper studies finite-sum optimization via stochastic oracle, which is used for empirical risk minimization in machine learning. A new variant of the stochastic Polyak is proposed for SSM (SGD in the non-differentiable setting).  The key modification consists in clipping the gradient norm in the step-size update rule (not in the algorithm) as an alternative to upper-bounding the step-size, as usually done to derive convergence guarantees.

Theoretical guarantees are provided in the context of non-smooth convex optimization, assuming there exists a minimizer, and assuming the functions are Lipschitz continuous.
Convergence to a neighborhood of the solutions (but not to the solution itself) is derived for an averaged version of the iterates.

A study of the step-size with a specific momentum method (called IMA) is also considered, and last-iterate convergence is studied for this problem.

The method is empirically evaluated on non-smooth convex problems (phase retrieval) and one deep neural-network training task (which is not convex but attracts a lot of attention).

**Compliance With Llm Reviewing Policy:**

Affirmed.

**Final Justification:**

I have been convinced by the follow up discussion with the authors. They agreed that some corrections or clarifications are required, which I believe can be done for the camera-ready version. Therefore I have raised my score and recommend acceptance.

**Key Questions For Authors:**

* What is the value of $M$ in the deep learning experiments? How often is this bound used?
* In theorem 3.4 is $\langle \partial f(y) , y-x\rangle$ always single valued for all sub-gradients $g$ in $\partial f(y)$? This would be worth mentionning.
* In the appendix, the author show that taking $M=1$ seem to always be a good value, but this seem contradicted by the Phase-retrieval experiments (Figure 9), and also in the main paper (Figure 2). Could the author clarify this?

**Limitations:**

yes

**Strengths And Weaknesses:**

The presentation of the paper is clear and easy to follow, the discussion with respect to prior work is clearly stated. The paper is mathematically rigorous with sound proofs.


The contribution is fair but the modification proposed does not seem to be a major one. This is not a valid critism on its own (a small modification may have a major impact), but the resulting theoretical results obtained only guarantee convergence to a neighborhood of the solution, not to the solution itself (in the case of SSM).
The contribution would have been more significant if convergence to the solution (possibly using decreasing step-sizes) had been derived, but this is not possible with the modification proposed (which is honestly and clearly explained).

On the other-hand, a strength of the paper is that the proposed method does not require knowledge of the optimal values $f_i^\ast$ but only lower-bounds of those, which is a significant improvement as this one of the main drawback of Polyak step-sizes.


I believe the paper could benefit from further discussion on the $G$-Lipschitz assumption on the functions $(f_i)_i$. Indeed, while this assumption is becoming increasingly popular in the litterature, I believe it requires being discussed. In particular, in the smooth case it implies that gradients are uniformly bounded which does not hold for basic functions such as $x\mapsto x^2$. I suggest the author further discuss this in the paper. The paper would benefit of an explanation on the fact that $\Vert g_i \Vert$ is already upper-bounded and that their method enforces a lower-bound, hence completely bounding the subgradient.


Another related weakness is that for small values of the parameter $M$ the radius of the neighborhood reached scales like $G^2 / M$, this sets the question on setting the value of $M$.

In the deep learning experiments I did not find the value of $M$ used, it could be interested to see how often this value is reached, in the same fashion as what the authors did for the upper-bound of the SPSmax step-size. For example, would a method using $\frac{f_i (x^t) - l_i^\ast}{M}$ work well or is the sub-gradient norm really important? (Since it is anyway upper-bounded)?



Overall, the method is interesting and assumptions are more realistic than what is usually assumed for Polyak step-sizes. Unfortunately this comes at the cost of weaker theoretical guarantees (convergence to a neighborhood). The experiments are promising but the choice and the role of the hyper-parameter $M$ is not completely clear, despite experiments and figures that try to assess the impact of $M$.
I am therefore not completely convinced of the interest of the method in practice, this could be fixed by stronger guarantees or additionnal strong empirical evidence that the method clearly outperforms concurent methods, in particular without requering the fine-tuning of $M$.

---

> ### Author Rebuttal · Authors · 2026-03-31
>
> **We thank the reviewer for the excellent evaluation of our presentation and for acknowledging the mathematical rigor of our proofs and the significant improvement of not requiring knowledge of the optimal values $f_i^*$.**
>
> Below, we address the questions and concerns.
>
> **About the convergence in a neighborhood:** We appreciate the reviewer's acknowledgment that "a small modification may have a major impact." We believe this is precisely the case here. The neighbourhood term $O(\sqrt{\max\{G^2, M\}/M} \cdot \sigma^2)$ is not a limitation specific to our method as it is an inherent feature of *any* non-decreasing or variance-reduced stochastic method. Similar neighbourhood terms appears in classical SGD with constant step sizes (Garrigos & Gower, 2023), momentum variants (Sebbouh et al., 2021; Liu et al., 2020), and all prior Polyak-type analyses (Loizou et al., 2021; Oikonomou & Loizou, 2025; Orabona & D'Orazio, 2025). Removing it without assuming interpolation or other strong conditions requires either decreasing step sizes or variance-reduction techniques, which is beyond the scope of this work.
>
> **On the value of $M$:** In all DNN experiments, we use $M = 1.0$ as the default. Regarding how often the safeguard is active (i.e., $\|g_i^t\|^2 < M$): this is architecture and epoch-dependent. In general, early in training the gradient norms tend to be larger, so the Polyak step is used, later in training, as loss decreases and gradients shrink, the safeguard activates more frequently. Crucially, unlike SPS_max where the constant bound $\gamma_b$ is selected in 98.45% of iterations (Section 2.2), our safeguard acts *in the denominator*, so the step size remains adaptive even when the safeguard is active. We can include a detailed count of safeguard activation frequency in the camera-ready version, analogous to the analysis we performed for SPS_max in Section 2.2.
>
> **Regarding the alternative $\frac{f_i(x^t) - \ell_i^*}{M}$:** this is an interesting simplification. When $\|g_i^t\|^2 < M$, SPS_safe essentially becomes $\gamma_t=\frac{f_i(x^t) - \ell_i^*}{M}$, while when $\|g_i^t\|^2 \geq M$, the gradient norm participates. The key advantage of keeping $\max\{\|g_i^t\|^2, M\}$ (rather than just $M$) is that for large gradients, the step size automatically adjusts. Using only $M$ in the denominator would effectively reduce to a decreasing learning-rate schedule $\propto f_i(x^t)/M$, losing adaptivity to gradient magnitude. The subgradient norm is therefore important, as it is already upper-bounded by the Lipschitz constant $G$ and our safeguard adds the extra lower bound on the denominator.
>
> **Regarding the value of $M$ in DNNs vs. Phase Retrieval experiments:** We agree this requires clarification. In the deep-learning experiments (Appendix D, Figures 10-15), $M = 1.0$ is consistently the best or near-best choice across all architectures and datasets. For the convex Phase Retrieval experiments (Figures 2, 9), the situation is different because the scale of the problem is different. The recommendation $M = 1.0$ is specifically for DNN practitioners. For convex problems, the optimal $M$ depends on problem-specific constants, and the sensitivity analysis (Figure 9) shows the expected theoretical behavior: larger $M$ produces tighter neighborhoods. We will clarify this distinction in the updated manuscript.
>
> **On the $G$-Lipschitz assumption:** Thank you for the suggestion. The $G$-Lipschitz assumption implies that subgradients are uniformly bounded: $\|g_i^t\| \leq G$ for all $i, t$. This is natural for many non-smooth ML losses (e.g., hinge loss, absolute loss) but does not hold in general. Our safeguard effectively bounds the denominator of our step size from both sides: $\|g_i^t\| \leq G$ from the Lipschitz assumption and $\max\{\|g_i^t\|^2, M\} \geq M$ from the safeguard. We will add a discussion of this in the paper.
>
> **On the single-valuedness of $\langle \partial f(y), y - x \rangle$ in Theorem 3.4:** In Theorem 3.4, we use a specific subgradient $g_i^t \in \partial f_i(x^t)$ (the one sampled by the algorithm), not the entire subdifferential. We will mention this more explicitly.
>
> **If you agree that we managed to address all issues, please consider raising your mark to support our work. If you believe this is not the case, please let us know so that we have a chance to respond.**

---

> > ### Author Rebuttal · Reviewer_thtm · 2026-04-01
> >
> > I thank the authors for clearly discussing the points I raised. They mostly addressed my concerns, although I still have a few points that require further discussion:
> >
> > * **On the convergence** Although I agree that vanilla SGD with constant step-sizes does not converge, one can enforce a decay (e.g. $1/k$) in the step-size of vanilla SGD and the proof is relatively straightforward to adapt to obtain convergence results (see e.g., [1, Theorem 6.8]). Therefore I disagree that this is beyond the scope of this work, indeed, either (1) a similar decay can be enforced in your method and the proof can be easily adapted to deduce convergence, in which case I recommend adding it -- or -- (2) Your method cannot be adapted directly to guarantee convergence as in SGD in which case this is a drawback of the method (and of several adaptive methods) and it should be acknowledged as such (the method has other merits).
> > * **On the use of $\langle \partial f(y), y-x\rangle$**. I understand that the algorithm uses a single element $g^t_i\in \partial f_i(y)$, but this is precisely why it must be justified how the results and the proof use $\langle \partial f(y), y-x\rangle$, and the meaning it has. Indeed, in general $\langle g, y-x\rangle$ may take a different value for each $g\in\partial f(y)$ unless there is a property stating that the value is the same for all $g$ hence justifying the notation $\langle \partial f(y), y-x\rangle$. I believe this needs to be clarified as the value obtained may depend on the element $g^t_i$ sampled. I also checked the Lemma C.3 in the reference used by the authors, there, it seems that $g^t$ is used in the definition of $B_f$, not $\partial f$.
> >
> > I am waiting for the next round of discussion regarding these remaining points before revising my score.
> >
> > [1] Garrigos & Gower. Handbook of Convergence Theorems for (Stochastic) Gradient Methods, 2024.
> >
> > **Edit** Following the discussion below (and provided that the authors implement the chances discussed), I have raised my score.

---

> > > ### Author Response · Authors · 2026-04-02
> > >
> > > **We thank the reviewer for the follow-up and for engaging in the discussion.** We appreciate the follow-up questions and the willingness to raise your score. Let us provide further clarification on the two points raised.
> > >
> > >
> > > **On convergence with decaying step sizes:**
> > > We agree with the reviewer on providing further clarifications in the updated version of our work related to convergence to a neighbourhood. The reviewer mentioned that we can either select a decreasing version of our step-size to guarantee convergence to exact solution (potentially trivial), or if this is not possible, we should explain the reason behind it.
> > >
> > >
> > > In our results, simply making the parameter $M_t$ decreasing over time does not work. The issue is that the convergence guarantee contains the neighborhood term $\sqrt{\frac{\max\{G^2,M_t\}}{M_t}}\sigma^2$. When $M_t$ decreases, we eventually have $\max\{G^2,M_t\}=G^2$, so this term becomes $\sqrt{\frac{G^2}{M_t}}\sigma^2$. Hence, as $M_t$ gets smaller, the factor $\sqrt{G^2/M_t}$ goes to infinity, causing the neighborhood term to blow up. Thus, such a simple decreasing $M_t$ is not sufficient, and a more careful strategy is required. Based on that we cannot easily adapt the current proof to deduce convergence.
> > >
> > >
> > > We would be happy to include the above reasoning in the updated version of our work. In that way, the readers would be able to understand better the convergence guarantees of our work. Having said that, we should note that our results are important as they are the first efficient extension of stochastic Polyak Step-size in the non-smooth regime.
> > >
> > >
> > > In the recently introduced stochastic Polyak literature, we are only aware of two works that address decreasing Polyak step sizes: Orvieto et al. (2022) and Jiang et al. (2023). Both were published years after the original SPS paper (Loizou et al., 2021) and propose significantly different step-size rules (DecSPS, AdaSPS) rather than modifications of SPS_max, and require substantially different proof techniques. This shows that obtaining exact convergence with Polyak-type rules is a non-trivial research contribution in its own right, not a routine adaptation. We expect in the coming years to be able to obtain the counterparts of DecSPS and AdaSPS for our safeguarded ideas for solving non-smooth problems, but we believe that this is not a trivial task. It is in that sense we originally mentioned that this is beyond the scope of our work. We hope that the above point answers your question clearly related to decreasing step sizes, and we would be happy to include such a discussion and details in the camera-ready version of our work.
> > >
> > >
> > > **On the use of $\langle\partial f(y), y-x\rangle$.** Your point is valid. Thanks for raising this. The Bregman divergence $B_f(x, y) = f(x) - f(y) - \langle g, x - y \rangle$ depends on the specific subgradient $g \in \partial f(y)$. In our proofs, when we take the conditional expectation of the stochastic inequality, the quantity that appears is $g^t:= \mathbb{E}_i[g_i^t \mid x^t]$, which by Lemma 9.5 in Garrigos & Gower (2024) is a subgradient of $f$ at $x^t$. The Bregman divergence $B_f(x^{t-1}, x^t) = f(x^{t-1}) - f(x^t) - \langle g^t, x^{t-1} - x^t \rangle$ is then defined with respect to this particular subgradient, and is nonnegative by the subgradient inequality. In the revised version, we will replace every occurrence of $\partial f(x^t)$ (both in the theorem statements and in the proofs) with the subgradient $g^t = \mathbb{E}_i[g_i^t \mid x^t]$, and add a remark citing Lemma 9.5 to justify that $g^t \in \partial f(x^t)$. We emphasize that the mathematical content of the theorems and proofs is correct; the issue is purely notational. Finally, we note that the Bregman divergence terms were included for completeness: the main convergence guarantees follow directly from their non-negativity, giving for example $\mathbb{E}[f(x^T)-f(x^*)]\leq\text{LHS of Thm 3.4}\leq O(1/\sqrt{T})+O(\sigma^2)$.
> > >
> > >
> > > **If you agree that we managed to address all remaining issues, we would appreciate your support for our work. If any concerns remain, please let us know.**

---

### Official Review · Reviewer_JFX2 · 2026-03-10

**Soundness:** 3
**Presentation:** 3
**Significance:** 2
**Originality:** 2
**Overall Recommendation:** 4
**Confidence:** 2

**Summary:**

This paper studies convex nonsmooth finite-sum optimization problems $\min_{x\in\mathbb{R}^d}[f(x)=\frac{1}{n}\sum_{i=1}^n f_i(x)]$ and works on the stochastic subgradient method (SSM) with stochastic Polyak step-sizes (SPSs). The authors propose a safeguarded step-size rule for SSM and its momentum variant Iterative Moving Average (IMA) update rule. The proposed rule replaces the denominator in the classical Polyak step-size with a safeguarded term of the form $\max(||g_i^t||^2, M)$, which prevents the step-size from exploding when subgradients $g_i^t\in\partial f_i(x^t)$ become small.

**Compliance With Llm Reviewing Policy:**

Affirmed.

**Final Justification:**

Thanks the response from the authors. I maintain my score.

**Key Questions For Authors:**

1. This paper emphasizes removing the requirement of knowing $f_i(x^\*)$. However, in many common machine learning losses the optimal per-sample loss value is known as zero. Could the authors clarify in which practical settings this assumption is restrictive?

2. The convergence guarantees still include a noise-dependent neighborhood term proportional to $\sigma^2$ (Table 1). Could the authors elaborate on whether the proposed safeguard meaningfully changes the stochastic convergence behavior compared to existing SPS variants (Loizou et al., 2021; Garrigos et al., 2023; Orabona & D'Orazio, 2025)?

3. Recent work like Orabona & D'Orazio (2025) discusses instability and neighborhood convergence issues for Polyak step sizes (Section 6 therein). How does the proposed safeguarded rule address or relate to these negative results?

**Limitations:**

yes.

**Strengths And Weaknesses:**

Strengths:

The paper studies stochastic Polyak step-sizes (SPSs) for convex nonsmooth stochastic optimization and proposes a safeguarded variant for stochastic subgradient methods (SSMs) and its Iterative Moving Average (IMA) variant. The analysis appears technically sound and the convergence guarantees follow the standard stochastic convex optimization techniques. The paper is generally clearly written and provides theoretical guarantees for both averaged (Theorem 3.1 and 3.4) and last iterates (Theorem 3.5).

Weaknesses:

The main limitation concerns the novelty relative to existing SPS methods. The proposed method essentially replaces the denominator $||g_i^t||^2$ with $\max(||g_i^t||^2, M)$. While this modification may improve stability, it does not substantially change the algorithmic design or the theoretical convergence rate compared to existing SPS methods.

Additionally, the discussion on $\sigma^2=0$ case for machine learning applications in Section 3 appears somewhat limited, given that the $\sigma^2$-dependent term remains in Theorem 3.1, 3.4, and 3.5.

---

> ### Author Rebuttal · Authors · 2026-03-31
>
> **We thank the reviewer for their time and for recognizing that the analysis is technically sound, the paper is clearly written, and convergence guarantees are provided for both averaged and last iterates.**
>
> Below, we address the key questions.
>
> **On the novelty of the modification:** We respectfully disagree that the contribution reduces to a simple denominator substitution. The modification is what *enables* the theoretical result: it is precisely the safeguarded form $\max\{\|g_i^t\|^2, M\}$ that allows us to remove both the interpolation assumption and the requirement of knowing $f_i(x^\ast)$. Identifying this step-size rule was the challenging part, prior works (e.g., Loizou et al., 2021, Appendix C.1) noted that extending to the non-interpolated setting should be straightforward using their proof techniques, yet no such result was provided. In practice, extending to this case is far from routine: previous proofs expand $\|x^{t+1} - x^\ast\|^2$, use convexity, and then apply loose inequalities on the step size, which misses much of the Polyak structure. In our case, we reverse-engineered the step size so that its precise expression is exploited in the upper bound, and then use algebraic identities with CS and Jensen's inequality to conclude. Our proof techniques and step-size construction therefore differ substantially from prior analyses.
>
> **On the $\sigma^2=0$ discussion:** We agree that the discussion can be expanded. Recall that $\sigma^2 = (\mathbb{E}_i[(f_i(x^\ast) - \ell_i^\ast)^2])^{1/2}$. Under interpolation (i.e., when $f_i(x^\ast) = f_i^\ast:=\inf f_i$ and choosing $\ell_i^\ast=f_i^\ast$), we get $\sigma^2 = 0$ and our bounds reduce to exact $O(T^{-1/2})$ convergence with no neighbourhood, recovering Loizou et al. (2021) as a special case (Corollary 3.2). Outside interpolation, the $\sigma^2$ term is unavoidable for *any* non-decreasing or variance-reduced stochastic method, for example: in classical SGD (Garrigos & Gower, 2023), momentum variants (Sebbouh et al., 2021), and all other Polyak-type analyses (Loizou et al., 2021; Oikonomou & Loizou, 2025; Orabona & D'Orazio, 2025). We will expand this discussion in the updated manuscript.
>
> **Q1:** Note that the optimal per-sample loss $f_i^\ast$ is different than $f_i(x^*)$ in general. They are equal only when the interpolation assumption is satisfied. While it is true that for many common ML losses the optimal per-sample loss is known to be zero, this is precisely the interpolation assumption, which was already handled by Loizou et al. (2021). Our contribution is relevant for all settings where interpolation does *not* hold, which includes: under-parameterized models (SVMs, shallow networks), regularised objectives, noisy or mislabelled data.
>
> In these settings, the exact value $f_i(x^\ast)$ is unknown and practically uncomputable, whereas a lower bound $\ell_i^* \leq f_i^\ast$ (e.g., $\ell_i^* = 0$ for nonnegative losses) is trivially available. This is the key practical advantage of our approach.
>
> **Q2:** The safeguard $M$ does not improve the convergence *rate* in big-O terms, all methods in this setting achieve $O(T^{-1/2})$. The meaningful change is in the *assumptions*:
> - Loizou et al. (2021): requires interpolation (strong, often impractical).
> - Garrigos et al. (2023): requires knowledge of $f_i(x^*)$ (oracle information, practically unavailable).
>
> SPS_safe achieves the same rate while removing all of these restrictions. It never degenerates into a constant step size (by construction), and requires only a lower bound $\ell_i^*$ that is trivially available for nonnegative losses. The practical impact is demonstrated by consistent empirical gains in Section 4.
>
> **Q3:** Orabona & D'Orazio (2025) is very relevant and we can strengthen the comparison. The practical distinction is that SPSmax (the step size they investigate) relies on an explicit upper bound $\gamma_b$, while our experiments show that this upper bound is often selected in practice, especially with the smoothed rule, effectively turning the method into a scheduled learning-rate scheme. Our SPS_safe, by contrast, safeguards the denominator rather than the entire step size, thereby preserving Polyak-style adaptivity and leading to the clipped-SSM interpretation. Our rule is designed to address exactly the practical instability/degeneration issue that motivates those negative observations. Furthermore, the results of Orabona & D'Orazio (2025) hold only for SSM and cannot be trivially extended to momentum variants (e.g., IMA), which we also handle in our work.
>
> Regarding the instability and neighborhood-convergence issues they raise for Polyak step sizes, we do not yet have a definitive answer for our proposed step size. This interesting direction for future work.
>
> **If you agree that we managed to address all issues, please consider raising your mark to support our work. If you believe this is not the case, please let us know so that we have a chance to respond.**

---

> > ### Author Rebuttal · Reviewer_JFX2 · 2026-04-02
> >
> > Thanks for the response.

---

### Official Review · Reviewer_BCf3 · 2026-03-12

**Soundness:** 3
**Presentation:** 3
**Significance:** 3
**Originality:** 3
**Overall Recommendation:** 4
**Confidence:** 3

**Summary:**

The manuscript assess a general context of adaptive optimization methods, specifically focusing on the arguably challenging non-smooth regime. Overall, the submission's specific aspect pertains to the stabilization of the Stochastic Polyak Step-size (SPS). The authors propose a novel variant called Safeguarded SPS (SPS_safe ) for the Stochastic Subgradient Method (SSM) and its momentum variant (IMA). By introducing a safeguard threshold 𝑀 in the denominator of the step size, the method prevents the learning rate from exploding when subgradients become arbitrarily small. The authors prove an O(1/T) convergence rate to a neighborhood of the optimal solution for convex, Lipschitz, non-smooth objectives without relying on restrictive interpolation assumptions or requiring knowledge of the optimal loss values . The theoretical findings are supported by experiments on convex problems (SVM, Phase Retrieval) and deep learning benchmarks (ResNet on CIFAR-10/100).

**Compliance With Llm Reviewing Policy:**

Affirmed.

**Key Questions For Authors:**

See above

**Limitations:**

yes

**Strengths And Weaknesses:**

**Strengths**
1. **Theoretical Contribution:** The removal of the interpolation assumption and the need for oracle knowledge of \(f_i(x^*)\) is a significant step forward for Polyak-type step sizes in non-smooth optimization. The theoretical bounds (Theorems 3.1, 3.4, and 3.5) are rigorous and clearly presented.
2. **Connection to Gradient Clipping:** Proposition 2.1 establishes the equivalence between \(SPS_{safe}\) and the adaptive Clipped SSM. This provides a strong theoretical justification for why clipping-like mechanisms work well in practice, bridging a gap between adaptive step sizes and standard deep learning training heuristics.
3. **Empirical Validation:** The authors validate their claims across different regimes. The convex experiments mirror the theoretical setup, while the deep neural network (DNN) experiments demonstrate the practical viability of the method. The analysis of gradient norms (Figure 4) visualizes the "vanishing gradient" mitigation claimed in the text.
4.**Motivation and Writing:** The paper is well-structured. The transition from the limitations of prior works (like \(SPS_{max}\) collapsing to a constant step size) to the proposed solution is logically sound and easy to follow.

**Weaknesses and Areas for Improvement**
1. **The Trade-off of the Safeguard \(M\):** While the method eliminates the upper bound \(\gamma_b\) used in \(SPS_{max}\), it introduces a new hyperparameter \(M\). The theoretical neighborhood size in Theorem 3.1 scales with \(\max\{G^2, M\}\sigma^2 / M\). If \(M\) is chosen to be very large to shrink the neighborhood, the step size \(\gamma_t\) becomes very small (essentially scaling as \(1/M\)), which would severely slow down convergence. The paper would benefit from a deeper theoretical or empirical discussion on the optimal trade-off for selecting \(M\).
2. **Gap Between Theory and Deep Learning Practice:** As is common in optimization literature, there is a gap between the convex theory and the highly non-convex deep learning experiments. While the empirical results on ResNets are promising, the theoretical guarantees do not cover non-convex or weakly convex landscapes. Acknowledging this limitation more explicitly in the conclusion would strengthen the paper's objectivity.
3. **Baseline Comparisons in DNNs:** In the deep learning experiments (Section 4.2), the baselines include SGD, Adam, and \(SPS_{max}\). However, for image classification tasks with ResNets, AdamW or SGD with a well-tuned cosine annealing schedule are the standard state-of-the-art baselines. Comparing \(SPS_{safe}\) against a heavily tuned baseline (e.g., SGD + Cosine Decay) would provide a more realistic assessment of its practical competitiveness.
4. **Regarding Appendix E (Smoothing trick for \(M\)):** The introduction of the exponential moving average \(M_t\) in Appendix E is highly practical and interesting. However, it introduces a new hyperparameter \(\beta\). Does the theoretical convergence guarantee still hold if \(M\) is replaced by the dynamic sequence \(M_t\)? If not, please clarify this in the appendix.
5. **Clarification on Figure 1 vs. Section 4.2:** In Figure 1, the authors show that \(SPS_{safe}\) exhibits a smoothing behavior similar to "Smooth \(SPS_{max}\)". In the main DNN experiments (Figure 3), does SPS_{safe} use a fixed \(M\) or the smoothed \(M_t\) from Appendix E? This should be explicitly stated in the main text to ensure reproducibility.
6. **Minibatch Size Impact:** How does the batch size interact with the safeguard \(M\)? Since the variance \(\sigma^2\) is intrinsically linked to the stochasticity of the gradients, does a larger batch size allow for a smaller \(M\)? An empirical ablation on batch size would be a valuable addition to the appendix.

---

> ### Author Rebuttal · Authors · 2026-03-31
>
> **We thank the reviewer for the positive evaluation and for noting that the theoretical contribution, connection to gradient clipping, empirical validation, and paper structure are all strengths of our work.**
>
> Below, we address each weakness and question.
>
> **Q1:** We agree that $M$ induces a trade-off: a larger $M$ tightens the final error neighborhood while potentially slowing early progress, whereas a smaller $M$ accelerates initially but leads to a larger neighborhood. In our submission, we already include a sensitivity study in the convex setting (Figure 2) and in the deep-learning setting. In particular, we have:
> - **Convex experiments:** We sweep $M \in \{0.1, 1.0, 5.0, 10.0, 20.0, 50.0, 100.0\}$ (Appendix D, Figure 9).
> - **Deep-learning experiments:** We report maximum accuracy as a function of $M$ for $M \in \{0.001, 0.01, 0.1, 1.0, 10.0, 100.0\}$ across multiple datasets and architectures (Appendix D, Figures 10-15).
>
> These results consistently indicate that *$M = 1.0$ is a reliable default* for practitioners. Furthermore, in Appendix E, we propose an exponentially averaged $M_t = \beta M_{t-1} + (1-\beta)\|g_i^t\|^2$ with $\beta = 0.9$ that is tuning-free and works very well in practice (Tables 3-4), matching or outperforming the best fixed $M$.
>
> **Q2:** We agree and will acknowledge this gap more explicitly in the conclusion. Our theory covers convex Lipschitz objectives, the DNN experiments serve as empirical evidence that the convex non-smooth insights transfer to non-convex practice. This approach is aligned with recent work by Schaipp et al. (ICML 2025), which demonstrates that bounds for subgradient methods on Lipschitz (possibly nonsmooth) convex objectives can closely track the empirical loss curves of large nonconvex models. We treat the DNN experiments accordingly, i.e. as empirical validation, not as a claim of non-convex theory.
>
> **Q3.** Thank you for the suggestion. We can include comparisons against SGD + cosine annealing and AdamW in the camera-ready version to provide a more comprehensive baseline evaluation. However, we note that the scope of our experimental section is to compare our safeguarded step size against the constant step size SGD/IMA and previously proposed Polyak steps
>
> **Q4.** The current convergence proofs (Theorems 3.1, 3.4, 3.5) indeed rely on a fixed constant $M$. When $M$ is replaced by the data-dependent sequence $M_t$, the theoretical guarantees do not directly carry over, and a new analysis would be required. In the paper, we emphasise that the smoothed $M_t$ is presented as a *practical recommendation*. However, at this point we have the following preliminary result about varying $M_t$:
> ***Theorem:***
> Consider the iterates of SSM with step size $\gamma_t =\frac{f_i(x^t)-\ell_i^\ast}{\max\{\|g_i^t\|^2,M_t\}}$,where $g_i^t\in\partial f_i(x^t)$, and where $M_t$ may be random and may depend on the current sample. Assume there exist deterministic constants $0<m\leq M_{\max}$ such that $m\leq M_t\leq M_{\max}$ Then, $\mathbb{E}[f(\overline{x^T})-f(x^\ast)]\leq\frac{\sqrt{\max\{G^2,M_{\max}\}}\|x^0-x^\ast\|}{\sqrt{T}}+\sqrt{\frac{\max\{G^2,M_{\max}\}}{m}}\,\sigma^2$.
>
> This theorem provides a first step toward convergence guarantees for general varying sequences $M_t$. In particular, it establishes an $O(1/\sqrt{T})$ convergence rate up to a neighborhood, provided that $M_t$ is lower and upper bounded. Moreover, the specific choice of $M_t$ proposed in Appendix E is provably upper bounded by $G^2$, which follows by induction. The proof of this result will, of course, be included in the camera-ready version.
>
>
> **Q5:** In all main DNN experiments (Figures 3, 5-7), SPS_safe uses a **fixed** $M=1.0$. The smoothed $M_t$ from Appendix E is presented separately (Tables 3-4) to demonstrate its effectiveness as a tuning-free alternative. We will explicitly state this in the main text.
>
> **Q6:** This is an interesting question. The safeguard $M$ should be chosen based on the scale of the subgradients, not on the batch size. The role of $M$ is to prevent the denominator from collapsing when $\|g_i^t\|\to0$. We do not have a formal batch-size ablation at this stage, but we can add one in the camera-ready version.
>
> **If you agree that we managed to address all issues, please consider raising your mark to support our work. If you believe this is not the case, please let us know so that we have a chance to respond.**

---

> > ### Author Rebuttal · Reviewer_BCf3 · 2026-04-03
> >
> > Thanks for your response

---

### Official Review · Reviewer_XP92 · 2026-03-12

**Soundness:** 3
**Presentation:** 3
**Significance:** 2
**Originality:** 3
**Overall Recommendation:** 4
**Confidence:** 3

**Summary:**

This paper addresses stochastic optimization for non-smooth convex objectives. Existing Stochastic Polyak Step (SPS) methods require impractical assumptions. To handle these shortages, the authors propose a safeguarded SPS, which explicitly bounds the gradient norm in the denominator with a threshold $M$ rather than capping the whole step size. They extend this to a momentum variant. Theoretically, the authors establish an $\mathcal{O}(T^{-1/2})$ convergence rate to a neighborhood of the optimum without requesting interpolation or optimal loss values. Empirically, $\text{SPS}_{\text{safe}}$ prevents vanishing gradient collapse during deep neural network training and achieves competitive performance on standard benchmarks and synthetic non-smooth tasks.

**Compliance With Llm Reviewing Policy:**

Affirmed.

**Key Questions For Authors:**

*Q1:* The bounds in Theorems 3.1 and 3.4 show convergence to a neighborhood parameterized by the safeguard $M$ and variance $\sigma^2$. Have the authors investigated theoretically or empirically the mechanism of decaying $M$ over time to achieve exact convergence, similar to decaying learning rates in standard stochastic subgradient methods?

*Q2:* The theoretical analysis requires knowing the true loss $l_i^{\ast}$. In Section 2.2, the authors set $l_i^{\ast}=0$ for deep neural networks. Could you mathematically justify this choice for your specific loss functions, especially in regimes where exact interpolation does not perfectly hold? Or are there any approaches to adjust $l_i^{\ast}$ during the algorithm?

*Q3:* Appendix E introduces a moving average $M_t$ to reduce manual tuning, which performs well. However, the convergence proofs rely only on a fixed constant $M$. Do the theoretical convergence guarantees still hold when $M$ is replaced by the dynamic, data-dependent sequence $M_t$?

**Limitations:**

Yes

**Strengths And Weaknesses:**

*Soundness:*
The theoretical foundation is technically solid in general. However, a theoretical gap exists: the bounds strictly require convexity and known lower bounds $l_i^{\ast}$, yet the DNN experiments arbitrarily set $l_i^*=0$ without rigorous justification (Section 2.2).

*Presentation:*
The paper is well-structured, with Table 1 effectively highlighting the theoretical gaps filled by this work. However, from my prespective, the comparison with the work (Orabona & D'Orazio, 2025) in Section 2.1 is somewhat defensive and lacks a detailed mathematical comparison.

*Significance:*
Removing the strict interpolation assumption and the oracle dependence on $f_i(x^*)$ for Polyak step is theoretically important.

*Originality:*
The main contribution is establishing the algebraic equivalence between the safeguarded SPS and adaptive gradient clipping (Proposition 2.1). This reframing provides theoretical guarantees for an empirical trick. However, the novelty in algorithm design is relatively incremental.

*Overview:*
This paper proposes $\text{SPS}_{\text{safe}}$ that bounds the gradient norm in the denominator to avoid vanishing gradients. The authors also provide an Iterate Moving Average (IMA) momentum variant. While the theoretical equivalence to adaptive gradient clipping removes restrictive interpolation assumptions, the overall contribution remains relatively incremental. The method achieves some empirical gains over Adam, introduces a highly sensitive hyperparameter $M$, and relies on theoretically unjustified heuristics, such as setting $l_i^*=0$, in practical deep learning applications.

I am willing to raise my score if:

- A rigorous mathematical discussion for setting the loss lower bound $l_i^* = 0$
- A formal convergence guarantee for the dynamic smoothing sequence ($M_t$) proposed in Appendix E
- Numerically demonstrate that a specific problem setting where $\text{SPS}_{\text{safe}}$ outperforms parameter-free baselines like Adam

---

> ### Author Rebuttal · Authors · 2026-03-31
>
> **We thank the reviewer for their detailed comments and for recognising that the theoretical foundation is technically solid and that removing the interpolation assumption and oracle dependence is theoretically important.**
>
> **Q1:** This is an interesting direction that we are actively investigating, although we do not yet have a definitive answer. A naive choice of making $M_t$ simply decrease over time does not work. The issue is that the convergence guarantee contains the neighborhood term $\sqrt{\frac{\max\{G^2,M_t\}}{M_t}}\sigma^2$. When $M_t$ decreases, we eventually have $\max\{G^2,M_t\}=G^2$, so this term becomes $\sqrt{\frac{G^2}{M_t}}\sigma^2$. Hence, as $M_t$ gets smaller, the factor $\sqrt{G^2/M_t}$ goes to infinity, causing the neighborhood term to blow up. Thus, a simple decreasing $M_t$ is not sufficient, and a more careful strategy is required. We are currently exploring such approaches, including variance-reduction techniques.
>
> **Q2:** We want to emphasize that the knowledge of $\ell_i^\ast$ is much more relaxed than both $f_i^\ast = \inf f_i$ and $f_i(x^\ast)$ required by prior works. Setting $\ell_i^\ast = 0$ is rigorously justified whenever the loss $f_i$ is nonnegative, since $\ell_i^\ast$ only needs to be *any* lower bound on $f_i(x)$ for all $x$. In virtually all standard ML applications (cross-entropy loss, hinge loss, squared loss, etc.) the individual losses are nonnegative by construction, so $\ell_i^\ast = 0$ is a valid (and provably correct) lower bound. This is not a heuristic: our theory holds for any $\ell_i^\ast \leq f_i^\ast = \inf_x f_i(x)$, and $0 \leq f_i(x)$ for all $x$ and $i$ in our experiments. The only consequence of using a looser bound is that the variance constant $\sigma^2 = (E_i[(f_i(x^\ast) - \ell_i^\ast)^2])^{1/2}$ is larger, but the convergence guarantee remains valid.
>
> We note that prior works (e.g., Loizou et al., 2021) effectively used $f_i^* = 0$ in their DNN experiments as well, but without explicitly justifying why the infimum should equal zero. In our setting, this choice is rigorously justified since our theory only requires a lower bound, not the exact optimal value.
>
> **Q3:** The current convergence proofs (Theorems 3.1, 3.4, 3.5) indeed rely on a fixed constant $M$. When $M$ is replaced by the data-dependent sequence $M_t$, the theoretical guarantees do not directly carry over, and a new analysis would be required. In the paper, we emphasise that the smoothed $M_t$ is presented as a *practical recommendation*. However, at this point we have the following preliminary result about varying $M_t$:
>
> ***Theorem:***
> Consider the iterates of SSM with step size $\gamma_t =\frac{f_i(x^t)-\ell_i^\ast}{\max\{\|g_i^t\|^2,M_t\}}$,where $g_i^t\in\partial f_i(x^t)$, and where $M_t$ may be random and may depend on the current sample. Assume there exist deterministic constants $0<m\leq M_{\max}$ such that $m\leq M_t\leq M_{\max}$ Then, $\mathbb{E}[f(\overline{x^T})-f(x^\ast)]\leq\frac{\sqrt{\max\{G^2,M_{\max}\}}\|x^0-x^*\|}{\sqrt{T}}+\sqrt{\frac{\max\{G^2,M_{\max}\}}{m}}\,\sigma^2$.
>
> This theorem provides a first step toward convergence guarantees for general varying sequences $M_t$. In particular, it establishes an $O(1/\sqrt{T})$ convergence rate up to a neighborhood, provided that $M_t$ is lower and upper bounded. Moreover, the specific choice of $M_t$ proposed in Appendix E is provably upper bounded by $G^2$, which follows by induction. The proof of this result will, of course, be included in the camera-ready version.
>
> **Regarding the comparison with Adam:** We would like to emphasize that our paper already includes experimental settings in which the proposed SPS_safe outperforms Adam. In particular, this behavior can be seen in Figures 3 and 5-6, where SPS_safe achieves better performance than Adam on the corresponding tasks. Thus, SPS_safe is competitive with it and can in fact outperform it as reported in the paper.
>
> **If you agree that we managed to address all issues, please consider raising your mark to support our work. If you believe this is not the case, please let us know so that we have a chance to respond.**

---

> > ### Author Rebuttal · Reviewer_XP92 · 2026-04-03
> >
> > I thank the authors for their efforts in addressing these questions.

---

### Decision · Program_Chairs · 2026-04-30

**Decision:**

Accept (regular)

**Comment:**

This paper considers Stochastic Polyak Step Sizes with a "safeguard" in the denominator to prevent from overly large step sizes.

Convergence rate analysis in terms of converging to a neighborhood without requiring interpolation or knowledge of the optimal loss value are provided, and some empirical results on convex problems and deep learning benchmarks (ResNet on CIFAR-10/100) support the effectiveness of the proposed method.

This work received a unanimous recommendation for acceptance, and during the rebuttal, the authors have provided fairly satisfying responses to each of the reviewers. Therefore, this paper is recommended for acceptance.